# $\epsilon$-Softmax: Approximating One-Hot Vectors for Mitigating Label Noise

**Jialiang Wang**[1][*]     **Xiong Zhou**[1][*]     **Deming Zhai**[1]
**Junjun Jiang**[1]     **Xiangyang Ji**[2]     **Xianming Liu**[1][†]
[1]Faculty of Computing, Harbin Institute of Technology
[2]Department of Automation, Tsinghua University
cswjl@stu.hit.edu.cn[*], cszx@hit.edu.cn[*], csxm@hit.edu.cn[†]

## Abstract

Noisy labels pose a common challenge for training accurate deep neural networks. To mitigate label noise, prior studies have proposed various robust loss functions to achieve noise tolerance in the presence of label noise, particularly symmetric losses. However, they usually suffer from the underfitting issue due to the overly strict symmetric condition. In this work, we propose a simple yet effective approach for relaxing the symmetric condition, namely $\epsilon$-**softmax**, which simply modifies the outputs of the softmax layer to approximate one-hot vectors with a controllable error $\epsilon$. Essentially, $\epsilon$-**softmax** *not only acts as an alternative for the softmax layer, but also implicitly plays the crucial role in modifying the loss function.* We prove theoretically that $\epsilon$-**softmax** can achieve noise-tolerant learning with controllable excess risk bound for almost any loss function. Recognizing that $\epsilon$-**softmax**-enhanced losses may slightly reduce fitting ability on clean datasets, we further incorporate them with one symmetric loss, thereby achieving a better trade-off between robustness and effective learning. Extensive experiments demonstrate the superiority of our method in mitigating synthetic and real-world label noise. The code is available at https://github.com/cswjl/eps-softmax.

## 1 Introduction

In recent years, deep neural networks (DNNs) have achieved remarkable advancements across various machine learning tasks [1, 2]. Despite its significant success, the prevalence of noisy labels in real-world datasets is a pervasive issue, often stemming from human bias or a lack of relevant professional knowledge [2]. The application of supervised learning methods directly to data with noisy labels consistently results in a decline in model performance [3]. Moreover, the ability to generalize from weak learners plays a pivotal role in the alignment of large language models [4]. Consequently, the pursuit of noise-tolerant learning has emerged as a compelling and significant challenge within the domain of weakly supervised learning, garnering increased attention in recent years [5, 6, 7, 8].

The literature presents several strategies for remedying this issue, with the design of robust loss functions standing out as a particularly popular approach due to its simplicity and broad applicability. Some previous works [9, 10, 5] theoretically proved that a loss function is noise-tolerant to label noise under mild conditions if it is symmetric:

$$\sum_{k=1}^{K} L(f(\mathbf{x}), k) = C, \quad \forall \mathbf{x} \in \mathcal{X}, \forall f \in \mathcal{H} \tag{1.1}$$

where $k \in [K]$ is the label corresponding to each class, $C$ is a constant, and $\mathcal{H}$ is the hypothesis class.

Furthermore, Asymmetric Loss Functions (ALFs) [7] are proposed as an extension of symmetric losses, which are designed for clean-label-dominant noise. However, both symmetric and asymmetric

---

[*]Equal contribution     [†]Corresponding author

38th Conference on Neural Information Processing Systems (NeurIPS 2024).

losses, such as Mean Absolute Error (MAE) [5] and Asymmetric Unhinged Loss (AUL) [7], encounter the underfitting problem and prove challenging to optimize [5, 6, 7]. The fitting ability of existing symmetric loss functions is constrained by the overly strict symmetric condition in Equation 1.1 [7]. Some approaches aim to improve the classical symmetric loss MAE by incorporating the robustness of the MAE and the rapid convergence of the Cross Entropy (CE). Examples include Generalized Cross Entropy (GCE) [11], Symmetric Cross Entropy (SCE) [12], and Jensen-Shannon Divergence Loss (JS) [13]. However, these loss functions often mechanically select an intermediate value between the derivatives of CE and MAE, essentially representing a trade-off between fitting ability and robustness. This prompts a crucial question: *How can we simultaneously achieve both robustness and effective learning?*

Zhou et al. [14] proposed an alternative approach to achieve the symmetric condition, diverging from the development of a new robust loss function. By restricting the hypothesis class $\mathcal{H}$, which restricts the outputs of the prediction function $f$ to one-hot vectors, any loss function can inherently become symmetric, i.e., $\sum_{k=1}^{K} L(f(\mathbf{x}), k) = C, \forall \mathbf{x} \in \mathcal{X}, \forall L \in \mathcal{L}$. However, a notable challenge arises from the fact that directly mapping outputs to one-hot vectors constitutes a non-differentiable operation. Accordingly, the crux of the matter lies in formulating an effective method to constrain the outputs to one-hot vectors. Previous attempts, such as temperature-dependent softmax [14], sparseness constraint [15], sparse regularization [14], and variance enlargement [16], have aimed to approximate one-hot vectors through the application of regularization methods. Nevertheless, these methods lack predictability, fail to achieve a quantitative approximation to one-hot vectors, and exhibit limited effectiveness, particularly at higher noise rates. Up to this point, a reliable approach for rigorously enforcing one-hot vector outputs remains elusive. Addressing this gap continues to pose a significant challenge in realizing the symmetric condition.

In this paper, we present a simple yet effective and theoretically sound approach for approximating outputs to one-hot vectors, which we term $\epsilon$-**softmax**. This method serves as a valuable alternative to the conventional softmax function in mitigating label noise. The distinctive attribute of $\epsilon$-**softmax** lies in its guarantee to possess a controllable approximation error $\epsilon$ to one-hot vectors, thus achieving perfect constraint for the hypothesis class. This approach is universally applicable across diverse models and loss functions, as it only needs to implement a simple layer resembling softmax. Specifically, the process of applying our $\epsilon$-**softmax** is outlined as follows:

> **Step 1.** $\mathbf{p}(\cdot|\mathbf{x}) \leftarrow \mathbf{softmax}(h(\mathbf{x})),$
>
> **Step 2.** $p_t \leftarrow p_t + m, \text{ where } t = \arg \max_{k \in [K]} p_k$
>
> **Step 3.** $\mathbf{p}(\cdot|\mathbf{x}) \leftarrow \mathbf{p}(\cdot|\mathbf{x})/(m+1).$

Herein, $\mathbf{p}(\cdot|\mathbf{x})$ represents the prediction probabilities, $p_k$ denotes the $k$-th element of the vector $\mathbf{p}(\cdot|\mathbf{x})$, and $h(\mathbf{x})$ denotes the logits. Step **1** obtains the original predictions by the softmax function. Step **2** involves a hyperparameter $m \geq 0$ to amplify the maximum term in the predictions with a controllable approximation error to one-hot vectors. Step **3** performs a normalization to make predictions sum to one, which also reduces the values of non-maximum terms.

The above description underscores that $\epsilon$-**softmax** as a plug-and-play module applicable to any classifier incorporating a softmax layer. Through the adjustment of the parameter $m$, our approach allows for the quantitative approximation of output to one-hot vectors, and thus owns the ability for mitigating label noise in classification. The main contributions of our work are highlighted as follows:

- We propose a simple yet effective scheme, $\epsilon$-**softmax**, for mitigating label noise. This scheme operates as a plug-and-play module, seamlessly integrating with any classifier that incorporates a softmax layer through just two additional lines of code.

- We offer rigorous theoretical analyses, which indicate that $\epsilon$-**softmax** is capable of controllably approximating one-hot vectors. Consequently, $\epsilon$-**softmax**-enhanced loss functions can achieve noise-tolerant learning and Bayes optimal top-$k$ error.

- We develop practical loss functions that enhance noise-tolerant learning. These include integration with MAE, achieving a better trade-off between robustness and effective learning. Extensive experimental results demonstrate the superiority of our method.

## 2 Preliminary

**Problem Formulation.** In a typical supervised classification scenario, let $\mathcal{X} \subset \mathbb{R}^d$ represent the $d$-dimensional input space, and $\mathcal{Y} = [K] = \{1, 2, ..., K\}$ is the label space, where $K$ is the number of classes. We are provided with a labeled dataset $\mathcal{S} = \{(\mathbf{x}_n, y_n)\}_{n=1}^N$, where each $(\mathbf{x}_n, y_n)$ is drawn *i.i.d.* from an underlying distribution $\mathcal{D}$ over $\mathcal{X} \times \mathcal{Y}$. The classifier $f$ is a mapping from the sample space to the label space, the prediction label $\hat{y} = \arg\max_k f(\mathbf{x})_k$. Here, the prediction function $f : \mathcal{X} \to \Delta_K$ estimates the probability $\mathbf{p}(\cdot|\mathbf{x})$, and $\Delta_K = \{\mathbf{u} \in [0,1]^K : \mathbf{1}^\top \mathbf{u} = 1\}$ represents the probability simplex. Typically, the function $f$ is expressed as $f = \mathbf{softmax} \circ h$, where $h$ denotes the logits input to the softmax layer. In the context of deep learning, $h$ is commonly a neural network. The objective or loss function is defined as a measure of distance $L : \Delta_K \times \Delta_K \to \mathbb{R}$. For a classification problem, the loss function is characterized by $L(\mathbf{u}, \mathbf{e}_y)$, where $\mathbf{e}_y$ represents the one-hot vector with its $y$-th element set to 1. In this study, we consider the loss functional $\mathcal{L}$, where $\forall L \in \mathcal{L}$, $L(\mathbf{u}, \mathbf{v}) = \sum_{k=1}^K \ell(u_k, v_k)$ with a basic loss function $\ell$. For brevity, we slightly abuse notation by defining $L(\mathbf{u}, k) = L(\mathbf{u}, \mathbf{e}_k)$.

**Label Noise Model.** In the context of learning with noisy labels, the accessible training set is the noisy counterpart $\tilde{\mathcal{S}} = \{(\mathbf{x}_n, \tilde{y}_n)\}_{n=1}^N$ rather than the clean set $\mathcal{S}$. We characterize the noise corruption process as the flipping of the clean label of $\mathbf{x}$ into its noisy version $\tilde{y}$ with a probability denoted as $\eta_{\mathbf{x},\tilde{y}} = p(\tilde{y}|\mathbf{x}, y)$. $\eta_{\mathbf{x}} = \sum_{k \neq y} \eta_{\mathbf{x},k}$ denotes the noise rate for $\mathbf{x}$. Our focus is on two prevalent types of label noise [6, 7] :

– *Symmetric or uniform noise*: $\eta_{\mathbf{x},y} = 1 - \eta$ and $\eta_{\mathbf{x},k \neq y} = \frac{\eta}{K-1}$,

– *Asymmetric or class-conditional noise*: $\eta_{\mathbf{x},y} = 1 - \eta_y$ and $\sum_{k \neq y} \eta_{\mathbf{x},k} = \eta_y$,

where $\eta_{\mathbf{x}} = \eta$ for symmetric noise, $\eta_{\mathbf{x}} = \eta_y$ denotes the noise rate for the $y$-th class, and $\eta_{\mathbf{x},i}$ is not necessarily equal to $\eta_{\mathbf{x},j}$, $i \neq j$ for asymmetric noise.

We also empirically consider learning with human-annotated noisy labels.

**Expected Risk and Noise Tolerance.** In learning with clean labels, given a loss function $L \in \mathcal{L}$ and a prediction function $f$, the expected risk with respect to $f$ is defined as: $\mathcal{R}_L(f) = \mathbb{E}_{(\mathbf{x},y)\sim\mathcal{D}}[L(f(\mathbf{x}), y)]$. The objective is to learn an optimal classifier $f^*$ that minimizes the expected risk, i.e., $f^* \in \arg\min_{f \in \mathcal{F}} \mathcal{R}_L(f)$.

In the case of learning with noisy labels, the corresponding noisy expected risk with respect to $f$ is defined as:

$$\mathcal{R}_L^\eta(f) = \mathbb{E}_\mathcal{D}\Big[(1-\eta_\mathbf{x})L(f(\mathbf{x}), y) + \sum_{k\neq y} \eta_{\mathbf{x},k} L(f(\mathbf{x}), k)\Big], \tag{2.1}$$

where $\sum_{k\neq y} \eta_{\mathbf{x},k} L(f(\mathbf{x}), k)$ is the noisy part that usually poses challenges in training accurate DNNs.

A loss function $L$ is claimed to be *noise-tolerant* if the global minimizer $f_\eta^*$ of $\mathcal{R}_L^\eta(f)$ also minimizes $\mathcal{R}_L(f)$, that is, $f_\eta^* \in \arg\min_f \mathcal{R}_L(f)$.

**All-$k$ Consistency.** Consistency is an important property of a loss function. A standard consistency is for achieving Bayes optimal top-1 error. We consider much stronger consistency for achieving Bayes optimal top-$k$ error for any $k \in [K]$. To this end, we introduce some definitions about top-$k$ consistency [17, 8].

For any vector $\mathbf{f} \in \mathbb{R}^K$ , we let $r_k(\mathbf{f})$ denote a top-$k$ selector that selects the $k$ indices of the largest entries of $\mathbf{f}$ by breaking ties arbitrarily. Given a data $(\mathbf{x}, y)$, its top-$k$ error is defined as $\mathrm{err}_k(f, \mathbf{x}, y) = \mathbb{I}(y \notin r_k(f(\mathbf{x})))$. The goal of a classification algorithm under the top-$k$ error metric is to learn a predictor $f$ that minimizes the $\mathrm{err}_k$ expected risk: $\mathcal{R}_{\mathrm{err}_k}(f) = \mathbb{E}_{(\mathbf{x},y)\sim\mathcal{D}}[\mathrm{err}_k(f, \mathbf{x}, y)]$.

For a fixed $k \in [K]$, a loss function $L$ is top-$k$ consistent if for any sequence of measurable functions $f : \mathcal{X} \to \Delta_K$, we have the global minimizer $f^*$ of $\mathcal{R}_L(f)$ also minimizes $\mathcal{R}_{\mathrm{err}_k}(f)$, that is, $f^* \in \arg\min_f \mathcal{R}_{\mathrm{err}_k}(f)$. If the above holds for all $k \in [K]$, it is referred to as *All-k consistency*.

# 3 Methodology and Theoretical Investigation

The symmetry condition in Equation 1.1, theoretically ensures that a symmetric loss function can be noise-tolerant [5]. Existing methods primarily focus on designing new loss functions. Those derived based on this design principle exhibit drawbacks, such as being challenging to optimize [5, 6] and prone to encounter the gradient explosion problem [7]. In this work, we take an alternative approach by proposing to constrain the hypothesis class $\mathcal{H}$ such that any loss functions will be approximately symmetric thereby rendering them robust to label noise.

## 3.1 Robustness

We introduce $\epsilon$-**softmax** to make the output $f(\mathbf{x})$ approximate one-hot vectors. The implementation of $\epsilon$-**softmax** is easy to follow, as outlined in the gray box of the Introduction Section 1, requiring just two additional lines of code alongside the standard softmax layer. This underscores that $\epsilon$-**softmax** is a plug-and-play module applicable to any classifier that incorporates a softmax layer. In this following, we investigate in theory how $\epsilon$-**softmax** realizes the controllable approximation of outputs to one-hot vectors, thereby enhancing the noise tolerance of any loss function.

**Approximating One-Hot Vectors.** We first introduce the concept of $\epsilon$-relaxation for a hypothesis class and then prove $\epsilon$-**softmax** can strictly approximate outputs to one-hot vectors with a controllable error.

**Definition 1** ($\epsilon$-relaxation). *Given a fixed vector $\mathbf{v}$ and its permutation set $\mathcal{P}_{\mathbf{v}}$ [1], the $\epsilon$-relaxation of $\mathcal{P}_{\mathbf{v}}$ is defined as the hypothesis class $\mathcal{H}_{\mathbf{v},\epsilon}$, in which any hypothesis $f \in \mathcal{H}_{\mathbf{v},\epsilon}$ outputs vectors in the $\epsilon$-ball of $\mathcal{P}_{\mathbf{v}}$, i.e., $\mathcal{H}_{\mathbf{v},\epsilon} = \{f : \min_{\mathbf{u} \in \mathcal{P}_{\mathbf{v}}} \|f(\mathbf{x}) - \mathbf{u}\|_2 \leq \epsilon, \forall \mathbf{x}\}$.*

Without loss of generality, we consider $\mathbf{v}$ as a one-hot vector, which is common in machine learning, to facilitate the implementation and analysis. We then denote the permutation set of the one-hot vector as $\mathcal{P}_{\mathbf{e}_1}$, where all elements are also one-hot vectors. In accordance with Definition 1, we can further derive that:

**Lemma 1.** $\epsilon$-**softmax** *can achieve $\epsilon$-relaxation for one-hot vectors:*

$$\min_{\mathbf{u} \in \mathcal{P}_{\mathbf{e}_1}} \|f(\mathbf{x}) - \mathbf{u}\|_2 \leq \epsilon = \frac{\sqrt{1-1/K}}{m+1}, \tag{3.1}$$

*where $f(\mathbf{x}) = \epsilon\text{-}\mathbf{softmax} \circ h(\mathbf{x})$.*

Lemma 1 suggests that $\epsilon$-**softmax** effectively enables $f(\mathbf{x})$ to approximate one-hot vectors with a controllable error $\frac{\sqrt{1-1/K}}{m+1}$.

**Robustness Guarantee.** We then establish theoretical guarantees for the robustness in mitigating label noise, where the constrained hypothesis class $\mathcal{H}_{\mathbf{e}_1,\epsilon}$ is considered.

Zhou et al. [14] established the excess risk bound [18] under symmetric noise, which holds when outputs fall within an $\epsilon$-relaxation of a permutation set. We prove a more comprehensive conclusion by considering asymmetric noise, of which symmetric noise is a special case.

**Theorem 1** (Excess Risk Bound under Asymmetric Noise). *In a multi-class classification problem, if the loss function $L \in \mathcal{L}$ satisfies $|\sum_{k=1}^{K}(L(\mathbf{u}_1, k) - L(\mathbf{u}_2, k))| \leq \delta$ when $\|\mathbf{u}_1 - \mathbf{u}_2\|_2 \leq \epsilon$, and $\delta \to 0$ as $\epsilon \to 0$, then for asymmetric label noise $\eta_{\mathbf{x},k} < (1 - \eta_y), \forall k \neq y$, if $\mathcal{R}_L(f^*) = 0$, the excess risk bound for $f \in \mathcal{H}_{\mathbf{v},\epsilon}$ can be expressed as*

$$\mathcal{R}_L(f_\eta^*) \leq 2\delta + \frac{2c\delta}{a}, \tag{3.2}$$

*where $c = \mathbb{E}_{\mathcal{D}}(1 - \eta_y)$, $a = \min_{\mathbf{x},k}(1 - \eta_y - \eta_{\mathbf{x},k})$, $f_\eta^*$ and $f^*$ denote the global minimum of $\mathcal{R}_L^\eta(f)$ and $\mathcal{R}_L(f)$, respectively.*

Theorem 1 demonstrate that under mild conditions for symmetric and asymmetric label noise, any loss function can be made noise-tolerant when the function $f(\mathbf{x})$ increasingly approximates a permutation set $\mathcal{P}_{\mathbf{v}}$ (i.e., $\delta \to 0$ as $\epsilon \to 0$).

---

[1]For example, consider the vector $\mathbf{v} = [v_1, v_2]$, its permutation set is defined as $\mathcal{P}_{\mathbf{v}} = \{[v_1, v_2], [v_2, v_1]\}$.

**$\epsilon$-Softmax-Enhanced Loss Functions.** Lemma 1 enable $f(\mathbf{x}) = \epsilon\text{-}\mathbf{softmax} \circ h(\mathbf{x})$ in closely approximating a one-hot vector, aligns with the principle outlined in Theorem 1 within the framework of the hypothesis class $\mathcal{H}_{\mathbf{e}_1, \epsilon}$. Hence, $\epsilon\text{-}\mathbf{softmax}$ progressively enhances the noise tolerance of any loss function as the hyperparameter $m$ approaches infinity ($\epsilon \to 0$ as $m \to \infty$ and the discrepancy $\delta \to 0$).

In this paper we consider CE loss and Focal loss (FL) [19]. We combine them with $\epsilon\text{-}\mathbf{softmax}$, denoted as $CE_\epsilon$ and $FL_\epsilon$. $\epsilon\text{-}\mathbf{softmax}$ approach is effective in adapting them to become more resilient to noise, ensuring better performance in the presence of label noise.

## 3.2 Consistency

Fundamentally, $\epsilon\text{-}\mathbf{softmax}$ *not only acts as an alternative for the softmax layer, but also plays the crucial role in modifying the loss function*. Consistency is an important property of a loss function. A standard consistency is for achieving Bayes optimal top-1 error. We show much stronger consistency for achieving Bayes optimal top-$k$ error for any $k \in [K]$ of the CE loss when combined with $\epsilon\text{-}\mathbf{softmax}$. To establish the All-$k$ consistency, we first introduce some existing results of sufficient condition of top-$k$ consistency by top-$k$ calibration [17, 8].

Let $P_k(\mathbf{f}, \mathbf{q})$ denote that $\mathbf{f}$ is top-$k$ preserving with respect to the underlying label distribution $\mathbf{q}$, i.e., if for all $l \in [K], q_l > q_{[k+1]} \Rightarrow f_l > f_{[k+1]}$, and $q_l < q_{[k]} \Rightarrow f_l < f_{[k]}$. Here, $q_{[k]}$ denotes he $k$-th greatest entry of $\mathbf{q}$. For example, if $\mathbf{q} = [0.2, 0.4, 0.4]$, then $q_{[1]} = 0.4, q_{[2]} = 0.4, q_{[3]} = 0.2$.

**Definition 2** (All-$k$ calibrated). *For a fixed $k \in [K]$, a loss function $L$ is called top-$k$ calibrated if for all $\mathbf{q} \in \Delta_K$ it holds that:*

$$\inf_{f \in \mathbb{R}^K : \neg P_k(f, \mathbf{q})} \mathcal{R}_L(f) > \inf_{f \in \mathbb{R}^K} R_L(f). \tag{3.3}$$

*A loss function is called All-$k$ calibrated if the loss function $L$ is top-$k$ calibrated for all $k \in [K]$.*

Yang and Koyejo [17] demonstrate that suppose $L$ is a nonnegative top-$k$ calibrated loss function, then $L$ is top-$k$ consistent. Furthermore, Zhu et al. [8] show that if $f^* = \arg\min_f R_L(f)$ is rank preserving with respect to $\mathbf{q}$, then $L$ is All-$k$ calibrated. $\mathbf{f}$ is called rank preserving w.r.t $\mathbf{q}$, i.e., if for any pair $q_i < q_j$ it holds that $f_i < f_j$ .

Then we establish comprehensive All-$k$ consistency for $CE_\epsilon$ as follows:

**Lemma 2.** *For one-hot label $\mathbf{e}_y$, $CE_\epsilon$ is All-$k$ calibrated and All-$k$ consistency.*

**Theorem 2.** *For any label $\mathbf{q} \in \Delta_K$, let $y = \arg\max_{k \in [K]} q_k$ and $t = \arg\max_{k \in [K]} p_k$ , if $t = y$ and $q_y - \max_{k \neq y} q_k > \frac{m}{m+1}$, $CE_\epsilon$ is All-$k$ calibrated and All-$k$ consistency.*

Lemma 2 and Theorem 2 mean that $CE_\epsilon$ performs well not only on the top-1 prediction, but also on the top-$k$ predictions for any $k \in [K]$. We show the All-$k$ consistency property of different losses in Table 1, the consistency of other losses refer to [8].

Table 1: All-$k$ consistency between different loss functions.

| Loss | CE | MAE | NCE | GCE | SCE | AUL | AGCE | AEL | LDR-KL | $CE_\epsilon$ |
|---|---|---|---|---|---|---|---|---|---|---|
| All-$k$ Consistency | ✓ | ✗ | ✗ | ✓ | ✓ | ✗ | ✗ | ✗ | ✓ | ✓ |

## 3.3 Gradient Analysis of $\epsilon$-Softmax.

To provide a comprehensive understanding of $\epsilon\text{-}\mathbf{softmax}$ in mitigating label noise, we further analyze the gradient of the CE loss when combined with $\epsilon\text{-}\mathbf{softmax}$. The gradient of $L_{CE_\epsilon}(f(\mathbf{x}), y)$ with respect to the model $h(\mathbf{x})$ can be derived as follows:

$$\frac{\partial L_{CE_\epsilon}(f(\mathbf{x}), y)}{\partial h(\mathbf{x})} = \begin{cases} -\frac{1}{p_y + m} \cdot \frac{\partial p_y}{\partial h(\mathbf{x})}, & t = y \\ -\frac{1}{p_y} \cdot \frac{\partial p_y}{\partial h(\mathbf{x})}, & t \neq y \end{cases}, \tag{3.4}$$

where $f = \epsilon\text{-}\mathbf{softmax} \circ h$, $\mathbf{p}(\mathbf{x}) = \mathbf{softmax}(h(\mathbf{x}))$ denotes the probabilities by standard softmax, and $t = \arg\max_{k \in [K]} p_k$ is the class with the largest value in prediction probabilities.

**Remark.** The gradient in Equation 3.4 shows that $CE_\epsilon$ will be equivalent to the standard CE if the maximum prediction is not the target class (i.e., $t \neq y$), in which the division of $m + 1$ in probabilities

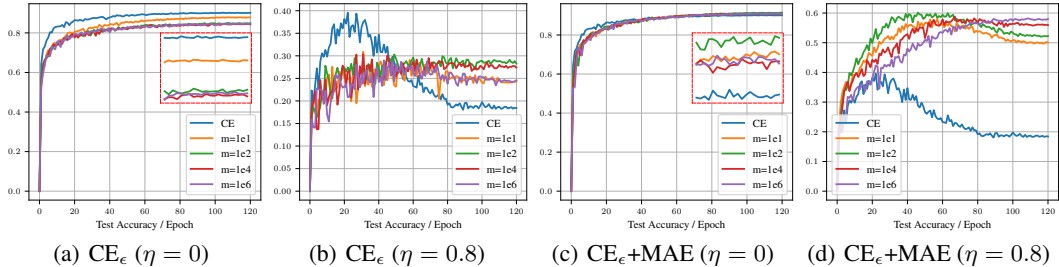

| (a) CE$_\epsilon$ ($\eta = 0$) | (b) CE$_\epsilon$ ($\eta = 0.8$) | (c) CE$_\epsilon$+MAE ($\eta = 0$) | (d) CE$_\epsilon$+MAE ($\eta = 0.8$) |

Figure 1: Test accuracies on CIFAR-10 under symmetric noise with different $m$, where the red box represents the zoomed-in accuracies of the last 20 epochs. (a) and (b) illustrate CE$_\epsilon$ with 0 (clean) and 0.8 noise rates, respectively. (c) and (d) illustrate CE$_\epsilon$+MAE ($\alpha = 0.01$, $\beta = 5$) similarly.

is omitted due to the partial deviation. Conversely, when the prediction class $t$ matches the target class $y$, the gradient undergoes dynamic scaling by $\frac{p_y}{p_y+m}$. This scaling results in smaller gradients, akin to a form of soft early-stopping [20], which facilitates the mitigation of overfitting to noisy labels. Such a characteristic enables Deep Neural Networks (DNNs) to efficiently fit clean samples in the early phases of training [21, 20], while simultaneously preventing the overfitting of noisy labels in the later stages of the training process. As illustrated in Figure 1(b), CE$_\epsilon$ achieves a stable test accuracy curve, even in the challenging scenario with 0.8 symmetric label noise, without overfitting to noisy labels. On the contrary, CE with the standard softmax tends to rapidly overfit to noisy labels after the early phase of training, leading to poor performance.

### 3.4 Better Trade-off between Robustness and Effective Learning

It can be noted that the incorporation of $\epsilon$-**softmax** somewhat sacrifices the fitting ability of the CE loss on clean datasets, as shown in Figure 1(a). Therefore, we need to enhance the fitting ability using additional techniques. Inspired by the Active Passive Loss [6], we propose to accommodate with the symmetric loss MAE. For instance, we formulate the combination of CE$_\epsilon$ and MAE (a.k.a., CE$_\epsilon$+MAE) as follows

$$L_{\text{CE}_\epsilon+\text{MAE}} = \alpha \cdot L_{\text{CE}_\epsilon} + \beta \cdot L_{\text{MAE}}, \tag{3.5}$$

ditto for FL$_\epsilon$+MAE.

**Lemma 3.** *For any loss function $L_\epsilon$ with $\epsilon$-**softmax** and symmetric loss function $L_{symmetric}$ defined in Equation 1.1, the excess risk bound of $\alpha \cdot L_\epsilon + \beta \cdot L_{symmetric}$ is equivalent to that of $\alpha \cdot L_\epsilon$.*

Lemma 3 suggests that the $\epsilon$-**softmax**-enhanced loss function $L_\epsilon$ can be seamlessly integrated with any symmetric loss function while not modifying the inherent robustness. As can be noticed in Figure 1(c) and Figure 1(d), CE$_\epsilon$+MAE not only depicts strong fitting capabilities but also achieves better noise tolerance. More interestingly, the test accuracy on clean datasets obtained by CE$_\epsilon$+MAE even exceeds that of the standard CE loss.

**Strict Convexity of CE$_\epsilon$+MAE.** To elaborate on how the combination of CE$_\epsilon$ and MAE can overcome the underfitting issue, we conduct an in-depth analysis from the optimization perspective. When the prediction $t = y$, the gradients of CE$_\epsilon$, CE and MAE w.r.t. $p_y \in (0, 1]$, are $-\frac{1}{p_y+m}$, $-\frac{1}{p_y}$ and $-2$, respectively. As can be seen, CE and CE$_\epsilon$ are strictly convex, while MAE exhibits linearity. Moreover, CE has stronger convexity compared to CE$_\epsilon$ (specifically, the gradient of CE changes more rapidly as $1/p_y^2 > 1/(p_y + m)^2$), rendering CE more susceptible to overfitting noisy labels while CE$_\epsilon$ suffering from underfitting for large $m$, as illustrated in Figure 1(a) and Figure 1(b). Conversely, owing to the linearity, MAE treats every sample equally, making it robust to label noise but leading to more training time for convergence [11]. Hence, the combination of CE$_\epsilon$ and MAE, which notably forms a strictly convex function (where the convexity can be controlled by $m$), can provide better trade-off between robustness and effective learning.

**Association with APL.** Additionally, our proposed CE$_\epsilon$+MAE coincides with the concept of active and passive losses in [6]. Specifically, for a loss function denoted as $L(f(\mathbf{x}), y) = \ell_1(f(\mathbf{x}), y) + \sum_{k \neq y} \ell_2(f(\mathbf{x}), k)$, $L$ is active if $\ell_2(f(\mathbf{x}), k) = 0$ for any $k \neq y$, and $L$ is passive if $\ell_2(f(\mathbf{x}), k) \neq 0$ for some $k \neq y$. Active losses only explicitly maximize the target probability $f(\mathbf{x})_y$, while passive losses also explicitly minimize non-target probabilities $\{f(\mathbf{x})_k\}_{k \neq y}$. For example, CE is an active loss, while MAE is passive. Based on these two loss terms, Ma et al. [6] proposed to combine a robust active loss and a robust passive loss into an "Active Passive Loss" (APL) framework for improving

Table 2: Last epoch test accuracies (%) of different methods on CIFAR-10/100 symmetric and asymmetric noise. The results "mean±std" are reported over 3 random runs and the top-2 best results are **boldfaced**.

| CIFAR-10 | Clean | Symmetric Noise Rate ($\eta$) | | | | Asymmetric Noise Rate ($\eta$) | | | |
|---|---|---|---|---|---|---|---|---|---|
| | | 0.2 | 0.4 | 0.6 | 0.8 | 0.1 | 0.2 | 0.3 | 0.4 |
| CE | 90.50±0.35 | 75.47±0.27 | 58.46±0.21 | 39.16±0.50 | 18.95±0.38 | 86.98±0.31 | 83.82±0.04 | 79.35±0.66 | 75.28±0.58 |
| FL | 89.70±0.24 | 74.50±0.18 | 58.23±0.40 | 38.69±0.06 | 19.47±0.74 | 86.64±0.12 | 83.08±0.07 | 79.34±0.30 | 74.68±0.31 |
| GCE | 89.42±0.21 | 86.87±0.06 | 82.24±0.25 | 68.43±0.26 | 25.82±1.03 | 88.43±0.20 | 86.17±0.29 | 80.72±0.42 | 74.01±0.53 |
| NLNL | 90.73±0.20 | 73.70±0.05 | 63.90±0.44 | 50.68±0.47 | 29.53±1.55 | 88.54±0.25 | 84.74±0.08 | 81.26±0.43 | 76.97±0.52 |
| SCE | 91.30±0.08 | 87.58±0.05 | 79.47±0.48 | 59.14±0.07 | 25.88±0.49 | 89.87±0.27 | 86.48±0.25 | 81.30±0.18 | 74.99±0.16 |
| NCE+MAE | 89.02±0.10 | 86.79±0.28 | 83.60±0.14 | 75.93±0.41 | 46.96±0.67 | 88.03±0.27 | 85.53±0.08 | 81.10±0.52 | 74.98±0.48 |
| NCE+RCE | 91.03±0.28 | 88.41±0.24 | 85.13±0.56 | 79.20±0.06 | 55.28±1.26 | 90.25±0.08 | 88.11±0.23 | 85.35±0.18 | **79.43±0.21** |
| NFL+RCE | 91.08±0.29 | 89.00±0.23 | 85.90±0.19 | 79.79±0.52 | 55.47±2.73 | 89.99±0.35 | 88.33±0.26 | 85.27±0.13 | 79.05±0.35 |
| NCE+AUL | 91.06±0.24 | 89.11±0.07 | 85.79±0.16 | 79.57±0.21 | 57.59±0.84 | 90.18±0.23 | 88.30±0.44 | 85.28±0.04 | 79.14±0.36 |
| NCE+AGCE | 91.13±0.11 | 89.00±0.29 | 85.91±0.15 | **80.36±0.36** | 49.98±4.81 | 89.90±0.09 | 88.36±0.11 | **85.73±0.12** | **79.28±0.37** |
| NCE+AEL | 88.43±0.25 | 86.46±0.28 | 83.06±0.23 | 75.15±0.32 | 43.22±0.46 | 87.59±0.38 | 85.98±0.14 | 82.87±0.16 | 75.78±0.12 |
| LDR-KL | 91.38±0.35 | 89.01±0.09 | 85.46±0.11 | 74.93±0.33 | 34.78±0.67 | 90.24±0.18 | 88.38±0.02 | 85.03±0.16 | 77.68±0.37 |
| CE+LC | 90.06±0.41 | 85.66±0.32 | 79.18±0.57 | 53.87±0.57 | 21.04±0.47 | 87.99±0.06 | 84.01±0.01 | 79.71±0.51 | 74.34±0.30 |
| $CE_\epsilon$+MAE | 91.40±0.12 | **89.29±0.10** | **85.93±0.19** | 79.52±0.14 | **58.96±0.70** | **90.30±0.11** | **88.62±0.18** | **85.56±0.12** | 78.91±0.25 |
| $FL_\epsilon$+MAE | 91.11±0.13 | **89.13±0.25** | **86.15±0.29** | **79.81±0.27** | **58.02±1.12** | **90.39±0.15** | **88.40±0.07** | 85.31±0.17 | 79.04±0.10 |

| CIFAR-100 | Clean | Symmetric Noise Rate ($\eta$) | | | | Asymmetric Noise Rate ($\eta$) | | | |
|---|---|---|---|---|---|---|---|---|---|
| | | 0.2 | 0.4 | 0.6 | 0.8 | 0.1 | 0.2 | 0.3 | 0.4 |
| CE | 70.79±0.58 | 56.21±2.04 | 39.31±0.74 | 22.38±0.74 | 7.33±0.10 | 65.10±0.74 | 58.26±0.31 | 49.99±0.54 | 41.15±1.04 |
| FL | 70.58±0.34 | 56.32±1.43 | 40.83±0.52 | 22.44±0.54 | 7.68±0.37 | 65.00±0.46 | 58.12±0.44 | 51.16±1.32 | 41.46±0.38 |
| GCE | 70.57±0.25 | 64.55±0.36 | 56.60±1.61 | 45.19±0.92 | 19.85±0.88 | 63.94±2.08 | 60.89±0.06 | 53.36±1.58 | 40.82±0.85 |
| NLNL | 68.72±0.60 | 46.99±0.91 | 30.29±1.64 | 16.60±0.90 | 11.01±2.48 | 59.55±1.22 | 50.19±0.56 | 42.81±1.13 | 35.10±0.20 |
| SCE | 70.41±0.20 | 55.23±0.76 | 40.23±0.29 | 21.44±0.52 | 7.63±0.24 | 64.54±0.30 | 57.62±0.70 | 50.17±0.19 | 41.01±0.74 |
| NCE+MAE | 67.69±0.05 | 63.21±0.44 | 57.91±0.45 | 45.26±0.44 | 23.72±0.99 | 65.70±1.04 | 62.87±0.42 | 55.82±0.19 | 41.86±0.27 |
| NCE+RCE | 67.89±0.47 | 64.60±0.92 | 58.64±0.19 | 45.25±0.50 | 24.87±0.52 | 66.20±0.28 | 63.18±0.37 | 55.05±0.32 | 41.21±0.66 |
| NFL+RCE | 68.28±0.30 | 64.57±0.52 | 57.64±0.74 | 45.47±0.59 | 24.35±0.32 | 66.18±0.38 | 63.63±0.30 | 55.33±0.25 | 40.82±0.67 |
| NCE+AUL | 69.55±0.40 | 65.12±0.36 | 55.86±0.20 | 37.88±0.32 | 12.69±0.14 | 67.06±0.23 | 58.16±0.17 | 48.06±0.16 | 38.30±0.12 |
| NCE+AGCE | 68.78±0.24 | 65.30±0.46 | **59.95±0.15** | 47.63±0.94 | 24.13±0.06 | 67.15±0.40 | 64.21±0.17 | 56.18±0.24 | 44.15±0.08 |
| NCE+AEL | 64.47±0.19 | 48.07±0.16 | 32.29±0.71 | 19.78±1.03 | 10.50±0.51 | 58.20±0.37 | 50.19±0.61 | 43.82±0.32 | 35.13±0.23 |
| LDR-KL | 71.03±0.28 | 56.69±0.06 | 40.69±0.66 | 22.59±0.23 | 7.49±0.33 | 65.93±0.01 | 58.47±0.04 | 50.92±0.15 | 41.94±0.37 |
| CE+LC | 71.80±0.34 | 56.26±0.09 | 37.36±0.49 | 17.46±0.62 | 6.32±0.16 | 65.85±0.30 | 58.84±0.02 | 50.46±0.12 | 40.97±0.39 |
| $CE_\epsilon$+MAE | 70.83±0.18 | **65.45±0.31** | 59.20±0.42 | **48.15±0.79** | **26.30±0.46** | **67.58±0.04** | **64.52±0.18** | **58.47±0.12** | **48.51±0.36** |
| $FL_\epsilon$+MAE | 70.58±0.68 | **65.45±1.39** | **59.58±0.80** | **48.09±0.35** | **26.73±0.45** | **67.73±0.12** | **64.80±0.29** | **58.88±0.30** | **48.10±0.23** |

sufficient learning with underfitting losses. Note that $CE_\epsilon$ is also active, thus $CE_\epsilon$+MAE coincides with the APL framework and further mitigates the underfitting issue.

To further validate $CE_\epsilon$+MAE, we incorporate it with sample selection, pseudo-label prediction [22], and MixUp [23], culminating in a semi-supervised learning algorithm we term $CE_\epsilon$+MAE (Semi). The algorithm details can be found in the Appendix C. In our experiments, we use "$CE_\epsilon$+MAE (Semi)" to ensure a fair comparison with other hybrid methods with sample selection and semi-supervised learning (SSL). No additional techniques are utilized for "$CE_\epsilon$+MAE".

## 4 Experiments

In this section, we conduct extensive experiments to validate the superiority of $\epsilon$-**softmax** in mitigating label noise. Complete experimental setting and results can be found in the Appendix D and E.

### 4.1 Evaluation on Benchmark Datasets

We evaluate our proposed methods on benchmark datasets CIFAR-10 / CIFAR-100 [24] with synthetic label noise, following [6, 7].

**Baselines.** We consider several baseline methods for comparison, including Standard CE and FL [19]; MAE; GCE [11]; NLNL [25]; SCE [12]; APL [6], including NCE+MAE, NCE+RCE, and NFL+RCE; AFLs [7], including NCE+AEL, NCE+AGCE, and NCE+AUL; LDR-KL [8]; and LogitClip [26], including CE+LC.

Table 3: Ablation experiments on CIFAR-100. The results "mean±std" are reported over 3 random runs and the best results are **boldfaced**. If $m = 0$, $CE_\epsilon$+MAE equals CE+MAE.

| CIFAR-100 | Clean | Symmetric | | Asymmetric |
| | | 0.4 | 0.8 | 0.4 |
| --- | --- | --- | --- | --- |
| CE | 70.79±0.58 | 39.31±0.74 | 7.33±0.10 | 41.15±1.04 |
| MAE | 5.31±1.19 | 2.78±1.68 | 2.13±0.98 | 3.11±0.26 |
| $CE_\epsilon$+MAE ($m = 0$) | 69.33±0.51 | 37.00±0.40 | 11.65±0.18 | 41.53±0.97 |
| $CE_\epsilon$+MAE ($m = 1e2$) | 70.55±0.47 | 39.39±0.77 | 13.05±0.58 | **48.51±0.36** |
| $CE_\epsilon$+MAE ($m = 1e4$) | 70.83±0.18 | **59.20±0.42** | **26.30±0.46** | 40.36±0.96 |
| $CE_\epsilon$+MAE ($m = 1e5$) | 67.72±0.88 | 56.41±0.22 | 22.14±0.56 | 7.56±1.10 |

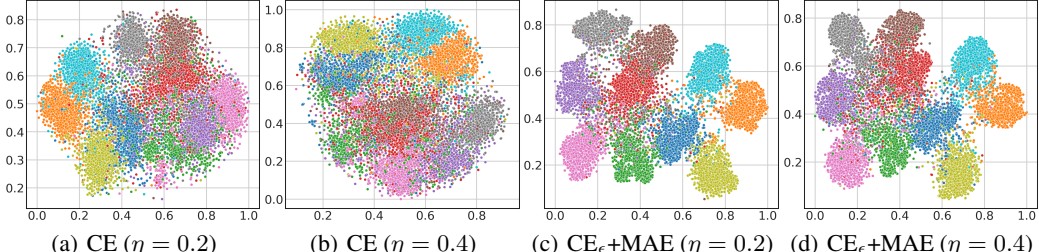

(a) CE ($\eta = 0.2$)     (b) CE ($\eta = 0.4$)     (c) $CE_\epsilon$+MAE ($\eta = 0.2$)     (d) $CE_\epsilon$+MAE ($\eta = 0.4$)

Figure 2: Visualizations of learned representations on CIFAR-10 with symmetric label noise. The x-axis and y-axis represent the first and second dimensions of the 2D embeddings, respectively.

**Results.** Table 2 presents the test accuracy of various loss functions under symmetric and asymmetric label noise. As can be seen, our proposed $\epsilon$-**softmax**-enhanced loss functions, $CE_\epsilon$+MAE and $FL_\epsilon$+MAE, demonstrate remarkable performance, ranking among the top-2 in most cases across both datasets. These methods consistently outperform others such as GCE, SCE, NLNL, NCE+MAE and LDR-KL, regardless of the noise rates. In scenarios of clean labels, $CE_\epsilon$+MAE and $FL_\epsilon$+MAE also exhibit strong fitting abilities, outperforming NCE+RCE and NCE+AGCE. In particular, on CIFAR-100 with 0.4 asymmetric noise, most robust loss functions have no effect, but our methods achieve over 48% accuracy, significantly outperforming all other methods. These findings underscore the robustness and effectiveness of $\epsilon$-**softmax**-enhanced loss functions, delivering their excellent performance in various noise scenarios.

**Ablation Experiments.** We perform detailed ablation experiments to further explore the role of each component and hyperparameter $m$ in our $CE_\epsilon$+MAE, experimental results are shown in Table 3. We can observe that CE will severely fit the noise label, and the symmetric loss MAE is difficult to optimize. CE+MAE (i.e., $m = 0$) is a trade-off between robustness and fitting ability, increasing noise tolerance at the cost of reducing fitting ability on clean labels, consistent with previous works [11, 12, 13]. In particular, our $CE_\epsilon$+MAE shows remarkable properties. As the parameter $m$ experiences a moderate increase, $CE_\epsilon$+MAE not only achieves noise tolerance for symmetric and asymmetric noise, but also achieves effective learning for the clean scenario. Additionally, the experimental results suggest that strict constraints are better suited for symmetric noise, while looser constraints are more effective for asymmetric noise.

**Visualization.** We conduct a further analysis to compare the effectiveness of $CE_\epsilon$+MAE and traditional CE in learning representations. We train models with different label noise and use the trained models to extract feature representations of the test set by t-SNE [27]. The visualizations for CIFAR-10 symmetric noise are depicted in Figure 2. Notably, the embeddings generated by CE show evident overfitting to label noise, as seen in the blending of embeddings from distinct classes. In sharp contrast, embeddings from the $CE_\epsilon$+MAE method consistently form clear, well-separated clusters, demonstrating its superior ability to learn robust and distinct representations under noisy label conditions.

## 4.2 Evaluation on Human-Annotated Datasets

We further conduct comparison studies on human-annotated datasets CIFAR-10N/CIFAR-100N [28], following the experiment setting in [28].

Table 4: Best epoch test accuracies (%) of different methods on CIFAR-N datasets. We compare methods without and with semi-supervised learning (SSL) and sample selection. The results "mean±std" are reported over 5 random runs and the best results are **boldfaced**.

| Method Without SSL | CIFAR-10N | | | | | CIFAR-100N |
|---|---|---|---|---|---|---|
| | Aggregate | Random 1 | Random 2 | Random 3 | Worst | Noisy |
| CE | 87.77±0.38 | 85.02±0.65 | 86.46±1.79 | 85.16±0.61 | 77.69±1.55 | 55.50±0.66 |
| Forward T | 88.24±0.22 | 86.88±0.50 | 86.14±0.24 | 87.04±0.35 | 79.79±0.46 | 57.01±1.03 |
| GCE | 87.85±0.70 | 87.61±0.28 | 87.70±0.56 | 87.58±0.29 | 80.66±0.35 | 56.73±0.30 |
| T-Revision | 88.52±0.17 | 88.33±0.32 | 87.71±1.02 | 87.79±0.67 | 80.48±1.20 | 51.55±0.31 |
| Peer Loss | 90.75±0.25 | 89.06±0.11 | 88.76±0.19 | 88.57±0.09 | 82.00±0.60 | 57.59±0.61 |
| F-Div | 91.64±0.34 | 89.70±0.40 | 89.79±0.12 | 89.55±0.49 | 82.53±0.52 | 57.10±0.65 |
| Negative-LS | **91.97±0.46** | 90.29±0.32 | 90.37±0.12 | 90.13±0.19 | 82.99±0.36 | 58.59±0.98 |
| VolMinNet | 89.70±0.21 | 88.30±0.12 | 88.27±0.09 | 88.19±0.41 | 80.53±0.20 | 57.80±0.31 |
| AGCE | 88.81±0.24 | 87.88±0.43 | 88.01±0.23 | 87.97±0.64 | 81.43±0.32 | N/A |
| $CE_\epsilon$+MAE | 91.80±0.33 | **90.43±0.29** | **90.53±0.28** | **90.64±0.35** | **83.74±0.43** | **61.78±0.14** |

| Method With SSL | CIFAR-10N | | | | | CIFAR-100N |
|---|---|---|---|---|---|---|
| | Aggregate | Random 1 | Random 2 | Random 3 | Worst | Noisy |
| Co-teaching+ | 90.61±0.22 | 89.70±0.27 | 89.47±0.18 | 89.54±0.22 | 83.26±0.17 | 57.88±0.24 |
| JoCoR | 91.44±0.05 | 90.30±0.20 | 90.21±0.19 | 90.11±0.21 | 83.37±0.30 | 59.97±0.24 |
| ELR+ | 94.83±0.10 | 94.43±0.41 | 94.20±0.24 | 94.34±0.22 | 91.09±1.60 | 66.72±0.07 |
| Divide-Mix | 95.01±0.71 | 95.16±0.19 | 95.23±0.07 | 95.21±0.14 | 92.56±0.42 | 71.13±0.48 |
| CORES* | 95.25±0.09 | 94.45±0.14 | 94.88±0.31 | 94.74±0.03 | 91.66±0.09 | 55.72±0.42 |
| CAL | 91.97±0.32 | 90.93±0.31 | 90.75±0.30 | 90.74±0.24 | 85.36±0.16 | 61.73±0.42 |
| PES (Semi) | 94.66±0.18 | 95.06±0.15 | 95.19±0.23 | 95.22±0.13 | 92.68±0.22 | 70.36±0.33 |
| SOP+ | 95.61±0.13 | 95.28±0.13 | 95.31±0.10 | 95.39±0.11 | 93.24±0.21 | 67.81±0.23 |
| Proto-semi | 95.03±0.14 | 95.48 ± 0.17 | 95.48±0.21 | 95.67±0.10 | 92.97±0.33 | 67.73±0.67 |
| $CE_\epsilon$+MAE (Semi) | **95.95±0.06** | **95.79±0.13** | **95.91±0.06** | **95.96±0.09** | **95.12±0.10** | **71.97±0.18** |

Table 5: Last epoch accuracies (%) on the WebVision and ILSVRC12 validation sets and the Clothing1M test set. The best results are **boldfaced**.

| Method | | CE | GCE | SCE | AGCE | NCE+RCE | NCE+AGCE | LDR-KL | $CE_\epsilon$+MAE |
|---|---|---|---|---|---|---|---|---|---|
| **WebVision** | Top-1 | 66.08 | 61.96 | 67.92 | 69.48 | 66.88 | 66.00 | 69.64 | **71.32** |
| | Top-5 | 84.76 | 76.80 | 86.36 | 87.28 | 86.48 | 85.20 | 87.16 | **88.48** |
| **ILSVRC12** | Top-1 | 60.72 | 60.52 | 63.28 | 65.12 | 63.96 | 62.68 | 65.24 | **67.20** |
| | Top-5 | 84.76 | 76.56 | 85.16 | 86.12 | 84.68 | 84.96 | 86.12 | **87.48** |
| **Clothing1M** | | 67.38 | 69.03 | 67.40 | 68.43 | 68.67 | 67.52 | 66.88 | **69.85** |

**Baselines.** For a fair comparison, we divide the baselines into those without and those with semi-supervised learning (SSL) and sample selection:

– *Without SSL*: Standard loss CE, Forward T [29], GCE [11], T-Revision [30], Peer Loss [31], F-Div [32], Negative-LS [33], VolMinNet [34], and AGCE [35].

– *With SSL*: Co-teaching+ [36], JoCoR [37], ELR+ [38], DivideMix [39], CORES* [40], CAL [41], PES (Semi) [20], SOP+ [42], and Proto-semi [43].

**Results.** Table 4 reports the test accuracy results of each method on the human-annotated datasets. The results show that the proposed $CE_\epsilon$+MAE and $CE_\epsilon$+MAE (Semi) provide significant improvements in handling human-annotated label noise, especially at high noise rates. Among the methods without SSL, $CE_\epsilon$+MAE stands out on the CIFAR-100N "Noisy" case as the only method to exceed 61% accuracy. Within the methods with SSL, $CE_\epsilon$+MAE (Semi) shows a pronounced superiority in all scenarios, especially in the most difficult CIFAR-10N "Worst" case and CIFAR-100N "Noisy" case. In the CIFAR-10N "Worst" case, $CE_\epsilon$+MAE (Semi) achieves an impressive accuracy rate of over 95%, significantly outperforming competing methods. These results underscore the effectiveness of the $\epsilon$-**softmax**-enhanced loss function in counteracting label noise for human-annotated scenarios.

### 4.3 Evaluation on the Real-World Datasets

We perform experiments on massively real-world noisy datasets, including WebVision [44], ILSVRC12 (ImageNet) [45] and Clothing1M [46], following the experiment setting in [7].

**Results.** In Table 5, we showcase the accuracies achieved on WebVision, ILSVRC12 and Clothing1M by various leading methods. Notably, our $CE_\epsilon$+MAE method outshines others, achieving the highest results on all real-world datasets. It surpasses CE by approximately 5.5% on WebVision and 6.5% on ILSVRC12. For Clothing1M, we finetune a pretrained ResNet-50, so the differences between the methods are relatively small, but our method still achieves the best accuracy. These results underline the robustness and efficacy of the $\epsilon$-**softmax**-enhanced loss function in real-world scenarios.

## 5    Conclusion

In this paper, we introduced $\epsilon$-**softmax**, a simple yet effective and theoretically sound scheme for noise-tolerant learning. Our method is not only easy to implement but also can be seamlessly integrated with any softmax-based DNNs, requiring just two additional lines of code. Our rigorous and comprehensive theoretical analysis reveals that $\epsilon$-**softmax** effectively alleviates the common issue of overfitting to noisy labels. Furthermore, we propose to incorporate $\epsilon$-**softmax**-enhanced loss functions with MAE, achieving better trade-off between effective learning and robustness. Extensive experimental results demonstrate the superior performance of our method in mitigating label noise.

## Broader Impacts

This work has the potential to advance the development of machine learning methods that can be deployed in contexts where it is costly to gather accurate annotations. This is an important issue in applications such as medicine, where machine learning has great potential societal impact. This work will not have negative social impacts.

## Acknowledgements

This work was supported by National Natural Science Foundation of China under Grants 92270116, 62071155 and 632B2031, and in part by the Fundamental Research Funds for the Central Universities (Grant No. HIT.DZJJ.2023075).

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

## A Limitation and Discussion

The limitation of $\epsilon$-**softmax** is that it may slightly reduce fitting ability on clean case. Therefore, we propose to combine the $\epsilon$-**softmax**-enhanced loss with the symmetric loss MAE. Consequently, our practical loss functions utilized for noise-tolerant learning exhibit a hybrid form similar to GCE and SCE, but their meanings are significantly different.

**Comparing with GCE and SCE.** GCE is a hybrid of CE and MAE var the negative Box-Cox transformation [11]. SCE combines CE with Reverse CE (RCE), where the RCE component actually acts as a scaled version of the MAE. This relationship is unveiled through the following derivation, adapted from Section 4.3 in SCE [12]: $L_{\mathrm{RCE}} = -\sum_{k=1}^{K} p(k \mid \mathbf{x}) \log q(k \mid \mathbf{x}) = -p(y \mid \mathbf{x}) \log 1 - \sum_{k \neq y} p(k \mid \mathbf{x}) A = -A \sum_{k \neq y} p(k \mid \mathbf{x}) = -A(1 - p(y \mid \mathbf{x})) = -\frac{A}{2} L_{\mathrm{MAE}}$. Consequently, SCE essentially translates to CE+MAE. Hence, GCE and SCE increases the fitting ability but reduces the robustness because of the CE term. Conversely, our $\mathrm{CE}_\epsilon$ is inherently robust. The combination of $\mathrm{CE}_\epsilon$ and MAE does not reduce the robustness, as demonstrated by Lemma 3, and also improves the fitting ability. We perform further experiments cimparing with GCE and CE+MAE (SCE), the results can be seen in Table 6. Our $\mathrm{CE}_\epsilon$+MAE obtains obviously the best results at all noise rates, significantly outperforming GCE and CE+MAE (SCE).

Meanwhile, we further compare our $\epsilon$-**softmax** with temperature-dependent softmax.

**Comparing with Temperature-Dependent Softmax.** $\mathbf{softmax}(\frac{h(x)}{\tau})$, where $\tau$ is the temperature parameter, is a useful technique for making outputs sparse [14]. Compared to our $\epsilon$-**softmax**, temperature-dependent softmax does not achieve a quantitative approximation to a one-hot vector for each output, and therefore cannot achieve a controllable excess risk bound. We also perform further experiments cimparing with temperature-dependent softmax. For simplicity, we refer to CE with temperature-dependent softmax as $\mathrm{CE}_\tau$, the results can be seen in Table 6. Our $\mathrm{CE}_\epsilon$+MAE obtains obviously the best results at all noise rates, significantly outperforming temperature-dependent softmax.

Table 6: Last epoch test accuracies (%) of ablation and comparetion experiments on CIFAR-100. The results "mean±std" are reported over 3 random runs. The best results are **boldfaced** and the best results of each method are underlined. If $m = 0$, $\mathrm{CE}_\epsilon$+MAE equals CE+MAE.

| CIFAR-100 | Clean | Symmetric 0.4 | 0.8 | Asymmetric 0.4 |
|---|---|---|---|---|
| CE | 70.79±0.58 | 39.31±0.74 | 7.33±0.10 | 41.15±1.04 |
| MAE | 5.31±1.19 | 2.78±1.68 | 2.13±0.98 | 3.11±0.26 |
| GCE ($q = 0.3$) | 70.31±0.95 | 38.72±0.87 | 6.43±0.17 | 38.79±1.47 |
| GCE ($q = 0.5$) | 70.57±0.25 | 50.61±0.64 | 8.16±0.40 | 38.58±0.55 |
| GCE ($q = 0.7$) | 65.22±1.57 | 56.60±1.61 | 18.23±0.25 | 40.82±0.85 |
| GCE ($q = 0.9$) | 18.27±2.43 | 17.61±2.25 | 19.85±0.88 | 13.96±1.69 |
| $\mathrm{CE}_\tau$+MAE ($\tau = 0.3$) | 70.00±1.51 | 36.87±2.12 | 14.61±0.47 | 40.37±3.10 |
| $\mathrm{CE}_\tau$+MAE ($\tau = 0.5$) | 69.57±0.46 | 47.99±0.48 | 13.62±0.24 | 45.53±1.19 |
| $\mathrm{CE}_\tau$+MAE ($\tau = 0.7$) | 70.11±0.71 | 36.08±2.21 | 10.58±0.20 | 46.92±0.45 |
| $\mathrm{CE}_\tau$+MAE ($\tau = 0.9$) | 69.32±0.27 | 36.34±1.47 | 11.19±0.04 | 42.27±0.92 |
| $\mathrm{CE}_\epsilon$+MAE ($m = 0$) | 69.33±0.51 | 39.72±0.67 | 11.65±0.18 | 41.53±0.97 |
| $\mathrm{CE}_\epsilon$+MAE ($m = 1e2$) | 70.55±0.47 | 48.39±0.53 | 13.05±0.58 | **48.51±0.36** |
| $\mathrm{CE}_\epsilon$+MAE ($m = 1e4$) | 70.83±0.18 | **59.20±0.42** | **26.30±0.46** | 40.36±0.96 |
| $\mathrm{CE}_\epsilon$+MAE ($m = 1e5$) | 67.72±0.88 | 54.99±1.05 | 22.14±0.56 | 7.56±1.10 |

## B Proof of Theorems

**Lemma 1.** $\epsilon$-**softmax** *can achieve $\epsilon$-relaxation for one-hot vectors:*

$$\min_{\mathbf{u} \in \mathcal{P}_{\mathbf{e}_1}} \| f(\mathbf{x}) - \mathbf{u} \|_2 \leq \epsilon = \frac{\sqrt{1 - 1/K}}{m + 1}, \tag{B.1}$$

*where $f(\mathbf{x}) = \epsilon\text{-}\mathbf{softmax} \circ h(\mathbf{x})$.*

*Proof.*

$$\min_{u \in \mathcal{P}_{\mathbf{e}_1}} \|f(\mathbf{x}) - \mathbf{u}\|_2 = \frac{\sqrt{1 - 2p_t + \sum_{k=1}^{K} p_k^2}}{m + 1}$$

$$= \frac{\sqrt{1 - p_t - \sum_{k=1}^{K} p_k(p_t - p_k)}}{m + 1}$$

$$\leq \frac{\sqrt{1 - p_t}}{m + 1} \leq \frac{\sqrt{1 - 1/K}}{m + 1}.$$

$\square$

**Theorem 1** (Excess Risk Bound under Asymmetric Noise). *In a multi-class classification problem, if the loss function $L \in \mathcal{L}$ satisfies $|\sum_{k=1}^{K}(L(\mathbf{u}_1, k) - L(\mathbf{u}_2, k))| \leq \delta$ when $\|\mathbf{u}_1 - \mathbf{u}_2\|_2 \leq \epsilon$, and $\delta \to 0$ as $\epsilon \to 0$, then for asymmetric label noise $\eta_{\mathbf{x},k} < (1 - \eta_y), \forall k \neq y$, if $\mathcal{R}_L(f^*) = 0$, the excess risk bound for $f \in \mathcal{H}_{\mathbf{v},\epsilon}$ can be expressed as*

$$\mathcal{R}_L(f_\eta^*) \leq 2\delta + \frac{2c\delta}{a}, \tag{B.2}$$

*where $c = \mathbb{E}_{\mathcal{D}}(1 - \eta_y)$, $a = \min_{\mathbf{x},k}(1 - \eta_y - \eta_{\mathbf{x},k})$, $f_\eta^*$ and $f^*$ denote the global minimum of $\mathcal{R}_L^\eta(f)$ and $\mathcal{R}_L(f)$, respectively.*

*Proof.*

$$R_L^\eta(f) = \mathbb{E}_{\mathcal{D}}\left[(1 - \eta_y) L(f(\mathbf{x}), y)\right] + \mathbb{E}_{\mathcal{D}}\left[\sum_{k \neq y} \eta_{\mathbf{x},k} L(f(\mathbf{x}), k)\right]$$

$$\leq \mathbb{E}_{\mathcal{D}}\left[(1 - \eta_y)\left(C + \delta - \sum_{k \neq y} L(f(\mathbf{x}), k)\right)\right] + \mathbb{E}_{\mathcal{D}}\left[\sum_{k \neq y} \eta_{\mathbf{x},k} L(f(\mathbf{x}), k)\right]$$

$$= (C + \delta)\mathbb{E}_{\mathcal{D}}(1 - \eta_y) - \mathbb{E}_{\mathcal{D}}\left[\sum_{k \neq y}(1 - \eta_y - \eta_{\mathbf{x},k}) L(f(\mathbf{x}), k)\right]$$

where $C = \sum_{k=1}^{K} L(\mathbf{v}, k)$, ditto

$$R_L^\eta(f) \geq (C - \delta)\mathbb{E}_{\mathcal{D}}(1 - \eta_y) - \mathbb{E}_{\mathcal{D}}\left[\sum_{k \neq y}(1 - \eta_y - \eta_{\mathbf{x},k}) L(f(\mathbf{x}), k)\right]$$

hence,

$$\left(R_L^\eta(f^*) - R_L^\eta(f_\eta^*)\right) \leq 2\delta \mathbb{E}_{\mathcal{D}}(1 - \eta_y) +$$

$$\mathbb{E}_{\mathcal{D}} \sum_{k \neq y}(1 - \eta_y - \eta_{\mathbf{x},k})\left[L(f_\eta^*(\mathbf{x}), k) - L(f^*(\mathbf{x}), k)\right]$$

According to the assumption $R_L(f^*) = 0$, we have $L(f^*(\mathbf{x}), y) = 0$ then $L(f^*(\mathbf{x}), k) = \frac{C}{k-1}$ where $k \neq y$. Since $L(f_\eta^*(\mathbf{x}), k) - L(f^*(\mathbf{x}), k) \leq 0$ where $k \neq y$, the second term on the right of the inequality is a non-positive value. And $R_L^\eta(f^*) - R_L^\eta(f_\eta^*) \geq 0$. So we have

$$\left| \mathbb{E}_{\mathcal{D}} \sum_{k \neq y}(1 - \eta_y - \eta_{\mathbf{x},k})\left(L(f_\eta^*(\mathbf{x}), k) - L(f^*(\mathbf{x}), k)\right) \right| \leq 2c\delta,$$

where $c = \mathbb{E}_{\mathcal{D}}(1 - \eta_y)$.

Let $a = \min_{\mathbf{x},k}(1 - \eta_y - \eta_{\mathbf{x},k})$, we have $\left| \mathbb{E}_{\mathcal{D}} \sum_{k \neq y}\left(L(f_\eta^*(\mathbf{x}), k) - L(f^*(\mathbf{x}), k)\right) \right| \leq \frac{2c\delta}{a}$. Note that $f_\eta^*, f^* \in \mathcal{H}_{\mathbf{v},\epsilon}$ means that $|\sum_k \left(L(f_\eta^*(\mathbf{x}), k) - L(f^*(\mathbf{x}), k)\right)| \leq 2\delta$, then we obtain

$$\left| \mathbb{E}_{\mathcal{D}}\left(L(f_\eta^*(\mathbf{x}), y) - L(f^*(\mathbf{x}), y)\right) \right| \leq 2\delta + \frac{2c\delta}{a},$$

that is, $\mathcal{R}_L(f_\eta^*) \le \mathcal{R}_L(f^*) + 2\delta + \frac{2c\delta}{a} = 2\delta + \frac{2c\delta}{a}$. □

**Lemma 2.** *For one-hot label* $\mathbf{e}_y$, $CE_\epsilon$ *is All-k calibrated and All-k consistency.*

*Proof.* Here $f = \epsilon\text{-}\mathbf{softmax} \circ h$, $\mathbf{p}(\cdot|\mathbf{x}) = \mathbf{softmax}(h(\mathbf{x}))$ denotes the probabilities by standard softmax, $p_k \in (0,1]$ and $t = \arg\max_{k \in [K]} p_k$ is the class with the largest value in prediction probabilities.

if $t = y$:

$$\frac{\partial L_{\text{CE}_\epsilon}(f(\mathbf{x}), y)}{\partial h(y|\mathbf{x})} = \frac{\partial - \log \frac{p_y + m}{m+1}}{\partial p_y} \cdot \frac{\partial p_y}{\partial h(y|\mathbf{x})} = -\frac{1}{m+1} \cdot \frac{m+1}{p_y + m} \cdot \frac{\partial p_y}{\partial h(y|\mathbf{x})}$$

$$= -\frac{1}{p_y + m} \cdot \frac{\partial p_y}{\partial h(y|\mathbf{x})} = -\frac{p_y}{p_y + m}(1 - p_y).$$

By the first-order optimality condition $\frac{\partial L_{\text{CE}_\epsilon}(f(\mathbf{x}), y)}{\partial h(\mathbf{x})} = 0$, we have: $p_y = 1$. Hence, for any $k \ne y$, we have $e_k = 0 < e_y$ and $p_k < p_y$.

if $t \ne y$:

$$\frac{\partial L_{\text{CE}_\epsilon}(f(\mathbf{x}), y)}{\partial h(y|\mathbf{x})} = \frac{\partial - \log \frac{p_y}{m+1}}{\partial p_y} \cdot \frac{\partial p_y}{\partial h(y|\mathbf{x})} = -\frac{1}{m+1} \cdot \frac{m+1}{p_y} \cdot \frac{\partial p_y}{\partial h(y|\mathbf{x})}$$

$$= -\frac{1}{p_y} \cdot \frac{\partial p_y}{\partial h(y|\mathbf{x})} = -(1 - p_y).$$

By the first-order optimality condition $\frac{\partial L_{\text{CE}_\epsilon}(f(\mathbf{x}), y)}{\partial h(\mathbf{x})} = 0$, we have: $p_y = 1$. Hence, for any $k \ne y$, we have $e_k = 0 < e_y$ and $p_k < p_y$.

Hence, $CE_\epsilon$ is All-$k$ calibrated. Since $CE_\epsilon$ is nonnegative, so $CE_\epsilon$ is All-$k$ consistency. □

**Theorem 2.** *For any label* $\mathbf{q} \in \Delta_K$, *let* $y = \arg\max_{k \in [K]} q_k$ *and* $t = \arg\max_{k \in [K]} p_k$, *if* $t = y$ *and* $q_y - \max_{k \ne y} q_k > \frac{m}{m+1}$, $CE_\epsilon$ *is All-k calibrated and All-k consistency.*

*Proof.* For $\frac{\partial L_{\text{CE}_\epsilon}(f(\mathbf{x}), \mathbf{q})}{\partial h(y|\mathbf{x})}$, we have:

$$\frac{\partial L_{\text{CE}_\epsilon}(f(\mathbf{x}), \mathbf{q})}{\partial h(y|\mathbf{x})} = -q_t \frac{m+1}{p_t + m} \cdot \frac{1}{m+1} \cdot \frac{\partial p_t}{\partial h(t|\mathbf{x})} - \sum_{k \ne t} q_k \frac{1}{p_k} \cdot \frac{\partial p_k}{\partial h(t|\mathbf{x})}$$

$$= -q_t \frac{1}{p_t + m} p_t(1 - p_t) - \sum_{k \ne t} q_k \frac{1}{p_k}(-p_k p_t).$$

By the first-order optimality condition $\frac{\partial L_{\text{CE}_\epsilon}(f(\mathbf{x}), \mathbf{q})}{\partial h(y|\mathbf{x})} = 0$, we have:

$$q_t \frac{1}{p_t + m} p_t(1 - p_t) = \sum_{k \ne t} q_k p_t$$

$$\Rightarrow \quad q_t \frac{1}{p_t + m}(1 - p_t) = 1 - q_t$$

$$\Rightarrow \quad p_t = q_t(1 + m) - m$$

Since, $\frac{m}{m+1} < q_t \le 1$, we can get $0 < p_t \le 1$.

For $\frac{\partial L_{\text{CE}_\epsilon}(f(\mathbf{x}), \mathbf{q})}{\partial h(j \ne y|\mathbf{x})}$, we have:

$$\frac{\partial L_{\text{CE}_\epsilon}(f(\mathbf{x}), \mathbf{q})}{\partial h(j \ne y|\mathbf{x})} = -q_t \frac{1}{p_t + m} \cdot \frac{\partial p_t}{\partial h(j|\mathbf{x})} - \sum_{k \ne t, j} q_k \frac{1}{p_k} \cdot \frac{\partial p_k}{\partial h(j|\mathbf{x})} - q_j \frac{1}{p_j} \cdot \frac{\partial p_j}{\partial h(j|\mathbf{x})}$$

$$= -q_t \frac{1}{p_t + m}(-p_j p_t) - \sum_{k \ne t, j} q_k \frac{q}{p_k}(-p_j p_k) + q_j(p_j - 1)$$

By the first-order optimality condition $\frac{\partial L_{CE_\epsilon}(f(\mathbf{x}),\mathbf{q})}{\partial h(j\neq y|\mathbf{x})} = 0$, we have:

$$q_t \frac{p_j p_t}{p_t + m} + \sum_{k\neq t,j} q_k p_j + q_j p_j = q_j$$

$$\Rightarrow \quad p_j = \frac{q_j}{\frac{q_t p_t}{p_t+m} + \sum_{k\neq t,j} q_k + q_j} = \frac{q_j}{\frac{q_t p_t}{p_t+m} + 1 - q_t}$$

Substituting $p_t = q_t(1+m) - m$, we can get $p_j = q_j(m+1)$. Since $q_t > \frac{m}{m+1}$, so $q_j < \frac{1}{m+1}$ and $0 < p_j < 1$.

For $i, j \neq t$, if $q_i < q_j$, we have $p_i < p_j$. Consider $q_{k\neq t}$ and $q_t$, because of the condition $q_y - q_{k\neq y} > \frac{m}{m+1}$, we have $q_k < q_t$, $q_t - q_k = q_t(1+m) - m - q_k(m+1) > 0$.

Hence, $CE_\epsilon$ is All-$k$ calibrated. Since $CE_\epsilon$ is nonnegative, so $CE_\epsilon$ is All-$k$ consistency. $\qquad\square$

**The gradient of $CE_\epsilon$.**

$$\frac{\partial L_{CE_\epsilon}(f(\mathbf{x}),y)}{\partial h(\mathbf{x})} = \begin{cases} -\frac{1}{p_y+m} \cdot \frac{\partial p_y}{\partial h(\mathbf{x})}, & t = y \\ -\frac{1}{p_y} \cdot \frac{\partial p_y}{\partial h(\mathbf{x})}, & t \neq y \end{cases}, \tag{B.3}$$

where $f = \epsilon\text{-}\mathbf{softmax} \circ h$, $\mathbf{p}(\mathbf{x}) = \mathbf{softmax}(h(\mathbf{x}))$ denotes the probabilities by standard softmax, and $t = \arg\max_{k\in[K]} p_k$ is the class with the largest value in prediction probabilities.

*Proof.* The proof is similar to Theorem 2.

if $t = y$:

$$\frac{\partial L_{CE_\epsilon}(f(\mathbf{x}),y)}{\partial h(\mathbf{x})} = \frac{\partial - \log \frac{p_y+m}{m+1}}{\partial p_y} \cdot \frac{\partial p_y}{\partial h(\mathbf{x})} = -\frac{1}{m+1} \cdot \frac{m+1}{p_y+m} \cdot \frac{\partial p_y}{\partial h(\mathbf{x})}$$

$$= -\frac{1}{p_y+m} \cdot \frac{\partial p_y}{\partial h(\mathbf{x})}.$$

if $t \neq y$:

$$\frac{\partial L_{CE_\epsilon}(f(\mathbf{x}),y)}{\partial h(\mathbf{x})} = \frac{\partial - \log \frac{p_y}{m+1}}{\partial p_y} \cdot \frac{\partial p_y}{\partial h(\mathbf{x})} = -\frac{1}{m+1} \cdot \frac{m+1}{p_y} \cdot \frac{\partial p_y}{\partial h(\mathbf{x})}$$

$$= -\frac{1}{p_y} \cdot \frac{\partial p_y}{\partial h(\mathbf{x})}.$$

$\qquad\square$

**Lemma 3.** *For any loss function $L_\epsilon$ with $\epsilon\text{-}\mathbf{softmax}$ and symmetric loss function $L_{symmetric}$ defined in Equation 1.1, the excess risk bound of $\alpha \cdot L_\epsilon + \beta \cdot L_{symmetric}$ is equivalent to that of $\alpha \cdot L_\epsilon$.*

*Proof.* For $\mathbf{u}_1, \mathbf{u}_2 \in \mathcal{H}_{\mathbf{v},\epsilon}$ and $\mathbf{u}_3, \mathbf{u}_4 \in \Delta_K$ , we have

$$|\sum_{k=1}^{K} (\alpha \cdot L_\epsilon(\mathbf{u}_1, k) + \beta \cdot L_{symmetric}(\mathbf{u}_3, k)) - \sum_{k=1}^{K} (\alpha \cdot L_\epsilon(\mathbf{u}_2, k) + \beta \cdot L_{symmetric}(\mathbf{u}_4, k))|$$

$$=|\sum_{k=1}^{K} \alpha \cdot L_\epsilon(\mathbf{u}_1, k) - \sum_{k=1}^{K} \alpha \cdot L_\epsilon(\mathbf{u}_2, k) + 0|$$

$$=\alpha \cdot |\sum_{k=1}^{K} \cdot L_\epsilon(\mathbf{u}_1, k) - \sum_{k=1}^{K} \cdot L_\epsilon(\mathbf{u}_2, k)|$$

$$\leq \alpha \cdot \delta$$

$\qquad\square$

## C The Algorithm of CE$_\epsilon$+MAE (Semi)

---

**Algorithm 1** CE$_\epsilon$+MAE (Semi)

---

1: **Input:** The noisy labeled dataset $\tilde{S} = \{(\mathbf{x}_n, \tilde{y}_n), n = 1, \cdots, N\}$, initialized model $f$, loss function $L_{\text{CE}_\epsilon+\text{MAE}}$, total epochs $T_{\text{all}}$, robust learning epochs $T_{\text{robust}}$ and trade-off parameter $\lambda$
2: **for** epoch $= 1$ to $T_{\text{robust}}$ do:
3:      Train $f$ on $\tilde{S}$ var $L_{\text{CE}_\epsilon+\text{MAE}}$
4: **end for**
5: **for** epoch $= T_{\text{robust}}$ to $T_{\text{all}}$ do:
6:      Sample selection: Divide the dataset $S$ into labeled (clean) dataset $S_l = \{(\mathbf{x}_n, y_n), n = 1, \cdots, |S_l|\}$ and unlabeled (noisy) dataset $S_u = \{(\mathbf{x}_n), n = 1, \cdots, |S_u|\}$
7:      **for** each minibatch $\mathcal{D}_l \in S_l$ and $\mathcal{D}_u \in S_u$ do:
8:          $q_n = \text{argmax}(f(x_n)), \quad x_n \in \mathcal{D}_u$    # Pseudo-label prediction
9:          $\hat{\mathcal{D}}_u\{(\hat{x}_n, q_n)\} = \text{Augment}(\mathcal{D}_u\{(x_n, q_n)\})$
10:         $\mathcal{W} = \text{shuffle}(\text{concat}(\mathcal{D}_l, \hat{\mathcal{D}}_u))$
11:         $\mathcal{D}_l' = \text{MixUp}(\mathcal{D}_l, \mathcal{W}_n) \quad n = 1, \cdots, |\mathcal{D}_l|$
12:         $\mathcal{D}_u' = \text{MixUp}(\mathcal{D}_u, \mathcal{W}_{n+|\mathcal{D}_l|}) \quad n = 1, \cdots, |\mathcal{D}_u|$
13:         $\text{Loss}_l = L_{\text{CE}_\epsilon+\text{MAE}}(f, \mathcal{D}_l')$
14:         $\text{Loss}_u = L_{\text{CE}_\epsilon+\text{MAE}}(f, \mathcal{D}_u')$
15:         $\text{Loss} = \text{Loss}_l + \lambda \cdot \text{Loss}_u$
16:         Train $f$ on Loss
17:      **end for**
18: **end for**
19: **return** $f$

---

**Algorithm Details and Parameters.** Reference to [20], we set $T_{\text{robust}} = 65$, $T_{\text{robust}} = 300$ and learning rate decay 0.1 at [60, 160, 260] epochs. Other experimental settings are the same as the CIFAR-N experiment [28] in the Appendix D.

For sample selection: We simply select $k$ samples from each class with the least loss as clean samples. For CIFAR-10N, we set $k = 2500$ for "Worst" case and 3500 for others. For CIFAR-100N, we set $k = 250$ for "Noisy" case and 350 for others. In practice, if $k > |\text{sample\_num}|$, we set $k = |\text{sample\_num}| - 20$.

For pseudo-label prediction: In the actual training, we do the pseudo-label prediction using two standard augment versions from the sample. We add the probabilities and divide by 2 to make the pseudo-label prediction. At the same time, we set the threshold $\sigma = 0.2$ and discard the samples whose prediction probability is less than the threshold.

For the Augment to $\mathcal{D}_u$, we employ RandAugment [47]. We set the trade-off parameter $\lambda$ to grow linearly from 0 to 1 over 200 epochs. The MixUp parameter $\alpha$ is set to 0.75 for epochs less than 100, and adjusted to 4 for epochs greater than 100. CE$_\epsilon$+MAE $m = 1e4, \alpha = 0.5, \beta = 1$ is the same as the CIFAR-N experiment for the robust learning stage and $m = 10, \alpha = 1, \beta = 1$ for the semi-supervised learning stage. In CE$_\epsilon$+MAE (Semi), we ensemble the outputs of two networks during inference and exchange the samples selected by the two networks during training, as is customary for methods that train two networks simultaneously [21, 36, 39, 38].

## D Experiments

### D.1 Evaluation on Benchmark Datasets

**Noise Generation.** We follow the approach of previous studies [6, 7] to experiment with two types of synthetic label noise: symmetric (uniform) noise and asymmetric (class-conditional) noise. In the case of symmetric label noise, we intentionally corrupt the training labels by randomly flipping labels within each class to incorrect labels in other classes. As for asymmetric label noise, we flip the labels

within a specific sets of classes: For CIFAR-10, the flips occur from TRUCK $\rightarrow$ AUTOMOBILE, BIRD $\rightarrow$ AIRPLANE, DEER $\rightarrow$ HORSE, and CAT $\leftrightarrow$ DOG. For CIFAR-100, the 100 classes are grouped into 20 super-classes, each containing 5 sub-classes, and we flip the labels within the same super-class into the next.

**Experimental Setting.** We follow the same experimental settings in [6, 7]: An 8-layer CNN is used for CIFAR-10 and a ResNet-34 for CIFAR-100. The networks are trained for 120 and 200 epochs for CIFAR-10 and CIFAR-100 with batch size 128. We use the SGD optimizer with momentum 0.9 and cosine learning rate annealing. The weight decay is set to $1 \times 10^{-4}$ and $1 \times 10^{-5}$ for CIFAR-10 and CIFAR-100. The initial learning rate is set to 0.01 for CIFAR-10 and 0.1 for CIFAR-100. Clipping the gradient norm to 5.0 and the minimum allowable value for $\log$ to $1 \times 10^{-8}$. Typical data augmentations including random shift and horizontal flip are applied to CIFAR-10; random shift, horizontal flip and random rotation are applied to CIFAR-100.

**Parameters Setting.** We set the parameter settings which match their original papers for all baseline methods [6, 7]. Specifically, for FL, we set $\gamma = 0.5$. For GCE, we set $q = 0.7$ for CIFAR-10, and $q = [0.5, 0.5, 0.7, 0.7, 0.9]$ for CIFAR-100 clean and symmetric noise ($\eta \in [0, 0.2, 0.4, 0.6, 0.8]$), $q = 0.7$ asymmetric noise. For SCE, we set $A = -4, \alpha = 0.1, \beta = 1$ for CIFAR-10, and $\alpha = 6, \beta = 0.1$ for CIFAR-100. For APL (NCE+MAE, NCE+RCE and NFL+RCE), we set $\alpha = 1, \beta = 1$ for CIFAR-10, and $\alpha = 10, \beta = 0.1$ for CIFAR-100. For NCE+AUL, we set $a = 6.3, q = 1.5, \alpha = 1, \beta = 4$ for CIFAR-10, and $a = 6, q = 3, \alpha = 10, \beta = 0.015$ for CIFAR-100. For NCE+AGCE, we set $a = 6, q = 1.5, \alpha = 1, \beta = 4$ for CIFAR-10, and $a = 1.8, q = 3, \alpha = 10, \beta = 0.1$ for CIFAR-100. For NCE+AEL, we set $a = 5, \alpha = 1, \beta = 4$ for CIFAR-10, and $a = 1.5, \alpha = 10, \beta = 0.1$ for CIFAR-100. For CE+LC, we set $\delta = [1, 1, 1, 1.5, 1.5]$ for CIFAR-10 clean and symmetric noise ($\eta \in [0, 0.2, 0.4, 0.6, 0.8]$) and $\delta = 2.5$ for CIFAR-10 asymmetric noise. We set $\delta = 2.5$ for CIFAR-100 asymmetric noise and $\delta = 0.5$ for others. For LDR-KL, We set $\lambda = 10$ for CIFAR-10 and 1 for CIFAR-100. For our $\text{CE}_\epsilon$+MAE, we set $\beta = 5, m = 1e5, \alpha = 0.01$ for CIFAR-10 symmetric, and $m = 1e3, \alpha = 0.02$ for asymmetric. For CIFAR-100, we set $\beta = 1$, $m = 1e4$ and $\alpha = [0.1, 0.05, 0.03, 0.0125, 0.0075]$ for clean and symmetric noise ($\eta \in [0, 0.2, 0.4, 0.6, 0.8]$), and $m = 1e2, \alpha = [0.015, 0.007, 0.005, 0.004]$ for asymmetric noise ($\eta \in [0.1, 0.2, 0.3, 0.4]$). For our $\text{FL}_\epsilon$+MAE, we set $\gamma = 0.1$ and others are same as $\text{CE}_\epsilon$+MAE. For NLNL, we use the results in [7] directly.

### D.2 Evaluation on Human-Annotated Datasets

**Experimental Setting.** We follow the experimental settings in [28]: Train a Resnet-34 using SGD for 100 epochs with initial learning rate 0.1, momentum 0.9, and weight decay 0.0005. Set the learning rate decay 0.1 at 60 epochs. Standard data augmentation including random shift and horizontal flip are applied. Best epoch test accuracies are compared. The results of the comparison methods are taken directly from [28] and the original papers [35, 43].

**Parameters Setting.** For our $\text{CE}_\epsilon$+MAE, we set $m = 1e4, \alpha = 0.5, \beta = 1$ for CIFAR-10N/100N. $\text{CE}_\epsilon$+MAE (Semi) has been covered in detail in the previous section C.

### D.3 Evaluation on Real-World Dataset WebVision

**Experimental Setting.** For WebVision, the training details follow [7]: We use the mini WebVision setting [6, 7] and train a ResNet-50 using SGD for 250 epochs with initial learning rate 0.4, nesterov momentum 0.9 and weight decay $3 \times 10^{-5}$ and batch size 256. The learning rate is multiplied by 0.97 after each epoch of training. All the images are resized to $224 \times 224$. Typical data augmentations including random width/height shift, color jittering, and horizontal flip are applied. We train the model on Webvision and evaluate the trained model on the same 50 concepts on the corresponding WebVision and ILSVRC12 validation sets.

For Clothing1M, we use ResNet-50 pre-trained on ImageNet similar to [46]. All the images are resized to $224 \times 224$. We use SGD with a momentum of 0.9, a weight decay of $1 \times 10^{-3}$, and batch size of 256. We train the network for 10 epochs with a learning rate of $5 \times 10^{-3}$ and a decay of 0.1 at 5 epochs. Typical data augmentations including random shift and horizontal flip are applied.

**Parameters Setting.** We set the best parameter settings which match their original papers for all baseline methods [6, 7]. Specifically, for GCE, we set $q = 0.7$ for WebVision and 0.6 for Clothing1M.

For SCE, we set $A = -4, \alpha = 10, \beta = 1$. For NCE+RCE, we set $\alpha = 50, \beta = 0.1$ for WebVision and $\alpha = 10, \beta = 1$ for Clothing1M. For AGCE, we set $a = 1e-5, q = 0.5$. For NCE+AGCE, we set $a = 2.5, q = 3, \alpha = 50, \beta = 0.1$. For LDR-KL, we set $\lambda = 1$. For our $CE_\epsilon$+MAE, we set $m = 1e3, \alpha = 0.015, \beta = 0.3$ for WebVison and $\alpha = 0.012, \beta = 0.1$ for Clothing1M.

## E   More Experimental Results

**Visualization.**   We show more visualizations of learned representations in Figure 3.

**Detailed Experimental Results of $CE_\epsilon$+MAE (Semi)**   The more detailed results are reported in Table 7.

**Instance-Dependent Noise.**   We follow the method in PDN [48] to generate instance-dependent noise. The experimental setting is the same as CIFAR-10/CIFAR-100. For $CE_\epsilon$+MAE on CIFAR-10, we set $\alpha = 0.045, \beta = 10, m = 1e5$. For CIFAR-100, we use the same parameters as symmetric noise. The results are reported in Table 8.

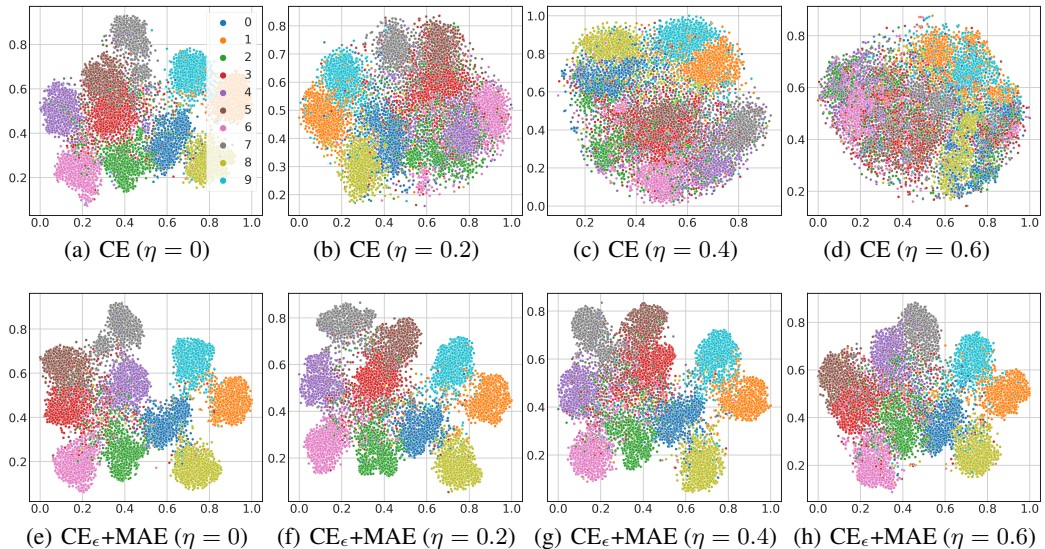

Figure 3: Visualizations of learned representations on CIFAR-10 with different symmetric label noise ($\eta \in [0, 0.2, 0.4, 0.6]$). The x-axis and y-axis represent the first and second dimensions of the 2D embeddings, respectively.

Table 7: Last and best epoch test accuracies (%) of $CE_\epsilon$+MAE (Semi) on CIFAR-N datasets. The results "mean±std" are reported over 5 random runs.

| $CE_\epsilon$+MAE (Semi) | CIFAR-10N | | | | | | CIFAR-100N | |
|---|---|---|---|---|---|---|---|---|
| | clean | Aggregate | Random 1 | Random 2 | Random 3 | Worst | clean | Noisy |
| Last | 96.06±0.15 | 95.83±0.14 | 95.76±0.12 | 95.83±0.12 | 95.87±0.11 | 95.01±0.16 | 78.54±0.33 | 71.78±0.23 |
| Best | 96.15±0.18 | 95.95±0.06 | 95.79±0.13 | 95.91±0.06 | 95.96±0.09 | 95.12±0.10 | 78.79±0.24 | 71.97±0.18 |

Table 8:  Last epoch test accuracies (%) on CIFAR-10/100 instance-dependent noise (IDN). The results "mean±std" are reported over 3 random runs and the best results are **boldfaced**.

| Method | CIFAR-10 IDN | | | CIFAR-100 IDN | | |
|---|---|---|---|---|---|---|
| | 0.2 | 0.4 | 0.6 | 0.2 | 0.4 | 0.6 |
| CE | 75.05±0.31 | 57.27±0.96 | 37.62±0.02 | 54.46±1.73 | 40.81±0.25 | 25.57±0.03 |
| GCE | 86.95±0.38 | 79.35±0.30 | 52.30±0.12 | 61.95±1.37 | 56.99±0.42 | 44.19±0.36 |
| SCE | 86.79±0.17 | 74.56±0.49 | 49.63±0.14 | 55.58±0.74 | 39.71±0.39 | 25.63±0.76 |
| NCE+RCE | 89.06±0.31 | 85.07±0.17 | 70.45±0.26 | 64.13±0.49 | 57.15±0.24 | 43.22±2.31 |
| NCE+AGCE | 88.90±0.22 | 85.16±0.26 | 72.68±0.21 | 65.33±0.18 | 58.59±0.68 | 43.42±0.24 |
| LDR-KL | 88.99±0.15 | 84.10±0.24 | 63.11±0.23 | 59.19±0.34 | 43.74±0.12 | 26.10±0.16 |
| $CE_\epsilon$+MAE | **89.27±0.42** | **85.26±0.29** | **74.32±0.89** | **67.44±0.19** | **60.80±0.20** | **46.53±0.54** |

