# OpenReview forum: "$\epsilon$-Softmax: Approximating One-Hot Vectors for Mitigating Label Noise"
_NeurIPS.cc/2024/Conference — NeurIPS 2024 poster_

### Official Review · Reviewer_4jLX · 2024-06-12

**Soundness:** 2
**Presentation:** 4
**Contribution:** 3
**Rating:** 5
**Confidence:** 5

**Summary:**

This paper proposes $\epsilon$-softmax to deal with label noise.  $\epsilon$-softmax modifies the outputs of the softmax layer to approximate one-hot vectors with a controllable error $\epsilon$. Both theoretical and empirical studies show the effectiveness of the proposed method.

**Strengths:**

1. The writing of this paper is good.
2. The robustness of the proposed loss is theoretically proved.

**Weaknesses:**

1. I think the motivation or the underlying reason for the effectiveness needs further explanation.
2. In experiment, the advantage of the proposed method over the competitors is probably not statistically significant.

**Questions:**

1. The term "symmetric condition" in abstract needs further explanation.
2. In the implementation steps in Line 61, is $\mathbf{p}(\cdot)$ the same as ${p}(\cdot)$? Or what is the relationship between these two notations? I think the mathematical notations should be strictly used and defined.
3. The robustness of the proposed $\epsilon$-softmax loss is theoretically justified. However, I'm curious to know the insight, or the underlying reason for its robustness. The authors wrote that "The distinctive attribute of $\epsilon$-softmax lies in its guarantee to possess a controllable approximation error $\epsilon$ to one-hot vectors, thus achieving perfect constraint for the hypothesis class." But, I cannot figure out why controlling approximation error $\epsilon$ to one-hot vectors and achieving perfect constraint for the hypothesis class are useful for handling label noise? What is the direct reason? It would be better if the authors can provide some intuitive explanations.
4. From the experiments, I note that the performance is a bit sensitive to different selection of m. Therefore, is it possible to give some guidance in choosing m for practical use? Besides, will better performance be obtained if we add the $\epsilon$-softmax loss functions with different m?
5. From the experimental results in Table 2, 4, I can see that the improvement of the proposed method over other methods is quite marginal. I guess if we do statistical significance test, maybe such improvement will not be statistically significant.

**Limitations:**

I think this work will not have negative social impacts.

---

> ### Author Rebuttal · Authors · 2024-08-07
>
> Thanks very much for your valuable comments. We would like to offer the following responses to your concerns.
>
> **1. Response to Weakness 1 and Question 3**
>
> Thanks for your kind comment.
>
> Previous work indicates that, for a fixed vector $\mathbf{v}$ and $\forall L \in \mathcal{L}$, we have
> $$\sum _{k=1}^K L(\mathbf{u}, k) = C, \forall \mathbf{u} \in \mathcal{P} _{\mathbf{v}},$$
>
> where $k \in [K]$ denotes the label corresponding to each class, $C$ is a constant. This lemma suggests that when the network output $\mathbf{u}$ is restricted to a permutation set $\mathcal{P}_{\mathbf{v}}$ of a fixed vector $\mathbf{v}$, any loss function will satisfy the symmetric condition.
>
> In this paper, we consider the fixed vector as a one-hot vector $\mathbf{e} _1$. The key challenge is that directly mapping outputs to a permutation set $\mathcal{P} _{\mathbf{v}}$ is a non-differentiable operation. Based on these motivations, we propose a simple yet effective scheme, $\epsilon$-softmax, to approximate one-hot vectors $\mathcal{P} _{\mathbf{e}_1}$. Thus, any loss function using $\epsilon$-softmax can mitigate label noise with a theoretical guarantee. We will include this detailed explanation in the revised version.
>
> **2. Response to Weakness 2 and Question 5**
>
> Thanks for your feedback.  We would like to emphasize that our experiments include dozens of  baselines and follow the same setting with previous works. On difficult noise and real world noise, our method achieves remarkable results.  These comprehensive evaluations aims to provide a robust assessment of our method.
>
> For Table 2, as suggested by other reviewer, we search more carefully about
> hyperparameters and obtain better performance. More details are available in the global rebuttal (see response to Q2). The new results are provided as follows, where "S" is symmetric noise,  "AS" is asymmetric noise, and * denotes using better learning rate and weight decay.
>
> |CIFAR-10|Clean|S (0.2)|S (0.4)|S (0.6)|S (0.8)|AS (0.1)|AS (0.2)|AS (0.3)|AS (0.4)|
> |:---:|:---:|:---:|:---:|:---:|:---:|:---:|:---:|:---:|:---:|
> |NCE+AGCE|91.13±0.11|89.00±0.29|85.91±0.15|80.36±0.36|49.98±4.81|89.90±0.09|88.36±0.11|85.73±0.12|79.28±0.37|
> |CE$_\epsilon$+MAE*|**91.94±0.18**|**89.76±0.08**|**86.77±0.16**|**80.47±0.87**|**58.96±0.70**|**91.06±0.13**|**89.58±0.29**|**87.78±0.23**|**82.47±0.56**|
> |**CIFAR-100**|**Clean**|**S (0.2)**|**S (0.4)**|**S (0.6)**|**S (0.8)**|**AS (0.1)**|**AS (0.2)**|**AS (0.3)**|**AS (0.4)**|
> |NCE+AGCE|68.78±0.24|65.30±0.46|59.95±0.15|47.63±0.94|24.13±0.06|67.15±0.40|64.21±0.17|56.18±0.24|44.15±0.08|
> |CE$_\epsilon$+MAE*|**75.62±0.23**|**70.96±0.20**|**64.22±0.50**|**50.69±0.25**|**26.30±0.46**|**72.86±0.06**|**66.70±0.20**|**58.47±0.12**|**48.51±0.36**|
>
> As can be seen, we significantly outperform the SOTA approaches.
>
> For Table 4, we used the t-test to determine if our CE$\epsilon$+MAE (Semi) statistically significantly exceeds the SOTA approaches. The p-values are as follows:
>
> |CIFAR-N|Aggregate|Random 1|Random 2|Random 3|Worst|Noisy100|
> |---|---|---|---|---|---|---|
> |Divide-Mix|0.018|0.000|0.000|0.000|0.000|0.006|
> |SOP+|0.001|0.000|0.000|0.000|0.000|0.000|
> |Proto-semi|0.000|0.012|0.002|0.001|0.000|0.000|
>
> As can be seen, in all cases, our approach statistically significantly exceeds the previous sota approaches (p-value $<$ 0.05).  This additional analyses will be included in the revised manuscript.
>
> **2. Response to Question 1**
>
> A loss function $L$  satisfies the symmetric condition is defined as (Eq. 1.1 in our Introduction):
>
> $$
> \sum_{k=1}^K L(f(\mathbf{x}), k) = C, \forall \mathbf{x} \in \mathcal{X}, \forall f \in \mathcal{H},
> $$
> where $k \in [K]$ denotes the label corresponding to each class, $C$ is a constant, and $\mathcal{H}$ is the hypothesis class. We will include this explanation in the abstract of the revised version to clarify this detail.
>
> **3. Response to Question 2**
>
> Thanks for your comment. $\mathbf{p(\cdot|x)}$ denotes the prediction probability **vector** for the sample $\mathbf{x}$. $p_k$ denotes the prediction probability **value** for the $k$-th class, i.e., the $k$-th element in the vector $\mathbf{p(\cdot|x)}$. We have followed common definitions of mathematical notations in academic paper, i.e., vectors are typically represented in boldface while scalars are usually represented in italicized font.
>
> **4. Response to Question 4**
>
> Thanks for your suggestions.
>
> --- Guidance on Choosing $m$
>
> An effective guideline is to use a larger $m$ for symmetric noise and a smaller $m$ for asymmetric noise. A larger $m$ imposes tighter symmetric constraints, which enhances robustness. However, if $m$ is too large, optimization on asymmetric noise can become challenging. For instance, on CIFAR-10, the symmetric MAE and CE achieve 45.36\% and 18.95\% accuracy under 0.8 symmetric noise, respectively. Conversely, MAE becomes difficult to optimize under asymmetric noise, resulting in 55.88\% and 75.28\% accuracy under 0.4 asymmetric noise.
>
> --- More experiments on different $m$
>
> We conducted further experiments on CIFAR-10 with various values of $m$. The results are summarized below. "S" is symmetric noise, and "AS" is asymmetric noise. We use m=1e5 for symmetric noise and 1e3 for asymmetric noise in the paper.
>
> |CIFAR-10|Clean|S (0.2)|S (0.4)|S (0.6)|S (0.8)|AS (0.1)|AS (0.2)|AS (0.3)|AS (0.4)|
> |---|:---:|:---:|:---:|:---:|:---:|:---:|:---:|:---:|:---:|
> |m=1e2|91.24±0.13|**89.33±0.09**|86.06±0.06|78.92±0.53|52.28±0.86|90.33±0.10|8.44±0.41 |85.16±0.18|78.36±0.16|
> |m=1e3|**91.49±0.22**|89.08±0.36|85.99±0.02|79.29±0.29|54.24±1.24|90.30±0.11|**88.62±0.18**|**85.56±0.12**|**78.91±0.25**|
> |m=1e4|91.02±0.21|89.32±0.40|**86.14±0.40**|**79.72±0.06**|57.41±1.32|90.20±0.04|88.49±0.39|82.73±4.26|71.65±0.66|
> |m=1e5|91.40±0.12|89.29±0.10|85.93±0.19|79.52±0.14|**58.96±0.70**|**90.44±0.10**|80.53±0.91|68.27±3.30|56.52±0.11|
>
> As shown above, using different values of $m$ can sometimes yield better performance, while the differences are not substantial.

---

> > ### Comment · Reviewer_4jLX · 2024-08-10
> > **Thanks for the rebuttal**
> >
> > Thanks for providing the additional results. I will increase my score. However, I still think the math should be rigorously defined and *explained* if the paper is finally accepted.

---

> > > ### Author Response · Authors · 2024-08-10
> > > **Thanks for your feedback**
> > >
> > > Thanks for your feedback! We are pleased to address your concerns. The strict definition and detailed explanation of the mathematical symbol are necessary. We will rigorously define the mathematical symbol in the final version. We truly appreciate the pivotal role that reviewers like yourself play in enhancing the quality of our work.

---

### Official Review · Reviewer_MvPF · 2024-07-08

**Soundness:** 3
**Presentation:** 3
**Contribution:** 3
**Rating:** 7
**Confidence:** 4

**Summary:**

This submission proposes a enhanced softmax layer for label-noise learning, namely $\epsilon$-softmax. By incorporating with the well-known $\epsilon$-relaxation, the proposed $\epsilon$-softmax can regularize the outputs of the model and avoid fitting the label-noise sample. This simple and plug-and-play method theoretically bounds the output logits to be an approximated one-hot vector. Extensive experiments demonstrate the effectiveness of the proposed method.

**Strengths:**

- The proposed method is simple, plug-and-play, and effective. Unlike other label-noise robust losses, the proposed method not only works well by itself, but also can be integrated with other label-noise robust method such as DivideMix. To the best of my knowledge, this could one of the first works endow such property.
- The theoretical analysis is comprehensive and make sense. The theoretical results suggests the proposed method possesses the Top-K error consistency and label-noise robustness.
- The analysis between the most related previous works is in reason. The basic idea that balances the label-noise robustness and learning effectiveness has been researched for a long time, e.g., GCE. This submission clearly presents the connection between the proposed method and other symmetric losses.
- The empirical results is effective and enough. This submission presents the comparison results between many label-noise robust losses and the proposed method achieves the best performance in most cases. Additionally, this submission provides the experimental results that demonstrated the plug-and-play property of the proposed method on sample-selection based method and loss-correction based method.

**Weaknesses:**

- The ablation studies on the gradient clipping should be conducted and providing experimental results with different backbones would be better.
- It is exhaustive and labor-expensive to find the optimal $m$ for diverse datasets.

**Questions:**

- Why choose the MAE as the loss to incorporate with ${CE_\epsilon}$? Following GCE and SCE, the MAE is indeed a practical and evaluated choice. Do there have any alternatives?
- What is the connection or relationship between logit adjustment and $\epsilon$-softmax?

**Limitations:**

No obvious limitations

---

> ### Author Rebuttal · Authors · 2024-08-07
>
> Thanks very much for your positive comments. We would like to offer the following responses to your concerns.
>
> **1. Response to Weakness 1**
>
> Thanks for you kind comment.
>
> --- About ablation study on gradient clipping
>
> We fully followed the experimental setup of previous work [1], and we used gradient clipping because it was used in [1].  This PyTorch component stabilizes training without impacting the method comparison. We report the results without using gradient clipping as follows,  where "S" is symmetric noise and "AS" is asymmetric noise.
>
> |CIFAR-10 w/o|Clean|S (0.2)|S (0.4)|S (0.6)|S (0.8)|AS (0.1)|AS (0.2)|AS (0.3)|AS (0.4)|
> |---|:---:|:---:|:---:|:---:|:---:|:---:|:---:|:---:|:---:|
> |CE|90.63±0.18|74.67±0.25|57.77±0.69|38.84±0.66|19.41±0.49|87.17±0.23|83.58±0.56|79.69±0.07|74.76±0.11|
> |NCE+AGCE|90.85±0.24|88.91±0.11|85.96±0.14|**79.97±0.12**|43.85±3.98 |90.13±0.09|88.40±0.16|85.27±0.36|**79.79±0.37**|
> |LDR-KL|91.07±0.11|89.15±0.12|84.98±0.52|74.77±0.31|32.20±0.78|90.22±0.27|88.50±0.47|85.43±0.17|77.86±0.07|
> |CE$\epsilon$+MAE|91.20±0.12|**89.22±0.25**|**86.10±0.21**|79.75±0.24 |**57.43±0.48**|**90.34±0.04**|**88.59±0.29**|**85.84±0.22**|78.72±1.09|
> |**CIFAR-100 w/o**|**Clean**|**S (0.2)**|**S (0.4)**|**S (0.6)**|**S (0.8)**|**AS (0.1)**|**AS (0.2)**|**AS (0.3)**|**AS (0.4)**|
> |CE|69.11±1.11|53.63±0.47|35.23±2.35|20.04±1.43|7.25±0.35|58.46±3.88|56.45±2.19|47.72±2.78|39.46±0.55|
> |NCE+AGCE|69.08±0.34|65.36±0.20|58.65±0.63|46.19±1.03|23.89±0.79|67.13±0.21|63.82±0.23|55.90±0.87|43.68±0.45|
> |LDR-KL|71.67±0.30|56.54±0.52|40.17±0.87|22.18±0.44|7.02±0.28|65.42±0.15|58.20±0.24|50.24±0.28|41.23±0.09|
> |CE$\epsilon$+MAE|70.20±0.78|**65.90±0.18**|**59.09±0.66**|**47.03±1.04**|**26.47±0.62**|**67.48±0.62**|**64.19±0.31**|**58.12±1.11**|**48.24±0.36**|
>
> As can be seen, whether gradient clipping is used or not does not affect the comparison of methods, and our method still shows excellent performance.
>
> --- About different backbone
>
> We conduct more experiments using VGG-13.
> The training settings are listed as follows: batch size 256, learning rate 0.1 with cosine annealing, weight decay 5e-4, and 200 epochs. The results with 3 random trials are as follows:
>
> |CIFAR-10|Clean|S (0.2)|S (0.4)|S (0.6)|S (0.8)|AS (0.1)|AS (0.2)|AS (0.3)|AS (0.4)|
> |---|:---:|:---:|:---:|:---:|:---:|:---:|:---:|:---:|:---:|
> |CE|93.91±0.32|83.21±0.08 |65.96±0.53 |43.30±0.60|21.46±0.82 |91.53±0.08|87.75±0.14|83.11±0.38|77.18±0.31|
> |GCE|93.90±0.09|91.15±0.19|82.57±0.21 |57.75±0.65|24.67±0.94|92.28±0.10|87.87±0.32|81.87±0.30|76.99±0.51 |
> |NCE+AGCE|90.42±0.09|88.24±0.33|83.61±0.86|50.63±5.62|11.68±2.92|89.98±0.14|88.03±0.45|86.46±0.53|69.02±0.45|
> |LDR-KL|92.51±0.13|90.53±0.19|87.91±0.03|79.24±0.13|39.49±0.58|91.64±0.17|89.86±0.35|87.50±0.40|67.63±0.24|
> |CE$\epsilon$+MAE|93.61±0.05|**91.62±0.24**|**88.38±0.39**|**80.31±0.47**|**48.05±1.50**|**92.67±0.38**|**90.13±0.19**|**87.99±0.07**|**81.69±0.48**|
> |**CIFAR-100**|**Clean**|**S (0.2)**|**S (0.4)**|**S (0.6)**|**S (0.8)**|**AS (0.1)**|**AS (0.2)**|**AS (0.3)**|**AS (0.4)**|
> |CE|71.71±0.18|59.96±0.47|47.26±0.40|30.50±0.14|9.68±0.53|66.62±0.59|60.47±0.33|53.35±1.09|43.26±0.24|
> |GCE|70.59±0.30|66.79±0.08|58.90±0.08|41.06±0.42|10.89±0.09|68.99±0.16|59.85±0.17|48.44±0.97|38.68±0.94|
> |NCE+AGCE|71.40±0.41|67.25±0.26|59.80±0.23|30.82±2.42|4.90±1.07|**69.85±0.01**|65.63±0.38|53.27±0.14|41.83±0.80|
> |LDR-KL|68.68±0.21|54.83±0.33|40.84±0.59|24.71±0.34|8.11±0.07|62.73±0.62|55.65±0.19|48.03±0.53|40.01±0.18|
> |CE$\epsilon$+MAE|71.07±0.21|**67.53±0.13**|**60.76±0.24**|**44.49±0.19**|**23.29±0.68**|69.28±0.09|**65.84±0.26**|**57.80±0.10**|**47.09±0.80**|
>
> As can be seen, our approach still has outstanding performance.
>
> **2. Response to Weakness 2**
>
> Thanks for your kind comment. Our method, like most robust loss functions, requires different parameters for different datasets. Many approachs have been proposed  to find optimal parameters quickly and economically. One efficient approach is to random select a subset of the training set as a smaller training set, for instance, 1/5 of the original set. We can then train on this smaller set to search for parameters efficiently.
>
> **3. Response to Question 1**
>
> Thanks for your insightful question. We chose MAE because it is the most classic symmetric loss, and there is no other reason. We recommend using any symmetric loss in combination with CE$_\epsilon$ rather than non-symmetric loss, because combining with symmetric loss does not increase the excess risk bound, as demonstrated by Lemma 2.
>
> **4. Response to Question 2**
>
> Thanks for your insightful question. Logit adjustment [2] modifies logits to encourage a large margin between rare and dominant labels for long-tail learning. However, logit adjustment requires prior knowledge of label distributions, which is not available for noisy labels. Therefore, it is not suitable for learning with noisy labels.
>
> $\epsilon$-Softmax mitigates overfitting to noisy labels by adjusting the probabilities. We find that $\epsilon$-softmax is also effective for long-tail learning, where we can use larger $m$ for dominant labels. Specifically, we set $m_k$ for the k-th class as $m \cdot p_k^{label}$, where $p_k^{label}$ is the frequency of the k-th class labels among all labels. We use ResNet-32 for CIFAR-10-Lt and CIFAR-100-LT, following the setting in [2] with long-tail imbalance ratio 100. The average class accuracies with 3 random trials are as follows:
>
> |Method|CIFAR-10-LT|CIFAR-100-LT|
> |---|---|---|
> |CE|70.21±0.34|39.18±0.27|
> |CE$_\epsilon$ (m=0.1)|71.75±0.46|40.08±0.25|
> |CE$_\epsilon$ (m=0.5)|**72.53±0.69**|40.32±0.06|
> |CE$_\epsilon$ (m=1)|72.50±0.40|**40.42±0.26**|
>
> Thanks again for your kind comment, which has inspired us to explore the application of $\epsilon$-softmax in other areas. We are pleased to see the flexibility of our plug-and-play method.
>
> [1] Asymmetric loss functions for noise-tolerant learning: Theory and applications, TPAMI, 2023.
>
> [2] Long-tail learning via logit adjustment, ICLR, 2021.

---

> > ### Comment · Reviewer_MvPF · 2024-08-10
> >
> > The authors have addressed my questions and concerns with additional results. I will keep my score.

---

> > > ### Author Response · Authors · 2024-08-11
> > > **Thanks for your feedback!**
> > >
> > > Dear Reviewer MvPF
> > >
> > > Thanks for your feedback! We are pleased to address your concerns and greatly appreciate your reviews, which play a crucial role in improving our work.
> > >
> > > Best regards,
> > >
> > > The authors

---

### Official Review · Reviewer_NkMU · 2024-07-09

**Soundness:** 2
**Presentation:** 2
**Contribution:** 2
**Rating:** 5
**Confidence:** 4

**Summary:**

This manuscript proposes a novel method to approximate the symmetric condition of the loss function, which is necessary for robustness to label noise. Specifically, the proposed method, named \\( \\epsilon \\)-softmax, can adjust the model output to approximate one-hot vector. However, the proposed method alone suffers from underfitting, so the authors combined it with MAE to achieve better performance. The authors evaluated the proposed method on datasets with different noise types and rates.

**Strengths:**

1. This manuscript proposes a novel and simple method to approximate the symmetric condition of loss function.
2. The theoretical analysis focuses not only on robustness to label noise but also on the top-k consistency of the loss function.
3. The proposed method was evaluated on various noise types and rates, including class-dependent noise and real-world noise.
4. It good to see the authors compare their proposed method with temperature-dependent softmax combined with MAE. The experimental results demonstrate the superiority of their proposed method compared to temperature-dependent softmax.

**Weaknesses:**

1. The theoretical discussion with temperature-dependent softmax is missing. As the authors mentioned in L42 to L53, there are other output restriction-based methods in the literature, such as temperature-dependent softmax. Although the authors claim that these methods “lack predictability, fail to achieve a quantitative approximation to one-hot vectors, and exhibit limited effectiveness,” there is no detailed discussion on why the proposed \\( \\epsilon \\)-softmax has superior properties.
2. A direct comparison with sparse regularization [1] is missing. Sparse regularization utilizes temperature- dependent softmax, which this manuscript has already compared, to approximate one-hot vector output. However, sparse regularization also employs an additional regularization term, \\( \\ell_p \\)-norm \\( \\| p(\\cdot | x) \\|^p_p \\) to enhance performance, and this regularization term is equivalent to MAE only if \\( p = 1 \\). It’s necessary to highlight the advantages of the proposed method compared to this highly relevant approach.
3. There is no ablation study on \\( \\alpha \\) and \\( \\beta \\). As the authors mentioned in L218, \\( \\epsilon \\)-softmax alone suffers from a loss in fitting ability, and they combined it with MAE to balance the robustness and effective learning. However, without the relevant ablation study, it’s unclear how this “trade-off” is achieved.
4. The theoretical discussions and experiments regarding instance-dependent label noise are overlooked. In recent years, the instance-dependent label noise has attracted increasing attention [2,3,4]. Experimenting the proposed method on instance-dependent label noise can provide a better understanding of how the proposed method performs with different types of label noise. I encourage the authors to include related discussion in the revised manuscript.

[1] Learning with noisy labels via sparse regularization, ICCV, 2021.

[2] Part-dependent Label Noise: Towards Instance-dependent Label Noise, NeurIPS, 2020.

[3] Learning with Instance-Dependent Label Noise: A Sample Sieve Approach, ICLR, 2021.

[4] Instance-Dependent Label-Noise Learning With Manifold-Regularized Transition Matrix Estimation, CVPR, 2022.

**Questions:**

1. Is \\( \\epsilon \\)-softmax + MAE still All-\\( k \\) calibrated and All-\\( k \\) consistency?
2. Can the proposed method perform better compared to sparse regularization?

**Limitations:**

The authors have acknowledged the limitations and societal impact of their work.

---

> ### Author Rebuttal · Authors · 2024-08-07
>
> Thanks very much for your positive comments. We would like to offer the following responses to your concerns.
>
> **1. Response to Weakness 1**
>
> Thanks for your insightful comment.
> In the following,  we  give the  theoretical discussion for temperature-dependent softmax.
>
> For model with temperature-dependent softmax, i.e., $f(\mathbf{x})= \tau$-softmax$ \circ h(\mathbf x)$, we have:
> $$\min _{\mathbf u \in \mathcal{P} _{\mathbf e_1}} \\|f(\mathbf x)-\mathbf u \\|_2=\sqrt{1-2p _{t}+\sum _{k=1}^K p_k^2} = \sqrt{1-p _{t}-\sum _{k=1}^K p_k(p_t-p_k)} \le \sqrt{1-p_t}\le \sqrt{1-1/K},$$
> where $\mathbf p(\cdot|\mathbf x) = f(\mathbf{x})$ and $t=\arg\max _{k\in[K]}p _k$. The equality holds when each term of $h(\mathbf x)$ is equal, no matter what value the temperature
>  $\tau$ takes, we have $p_k = 1/K, \forall k$.
>
> For $\epsilon$-softmax, we have $\min_{\mathbf u\in\mathcal{P}_{\mathbf e_1}}\\|f(\mathbf x)-\mathbf u\\|_2\le \epsilon = \tfrac{\sqrt{1 - 1/K}}{m+1}$ (Lemma 1). Our $\epsilon$-softmax make every output approximating one-hot vectors with a smaller error $\epsilon$. Thus, a smaller excess risk bound can be derived.
>
> **2. Response to Weakness 2 and Question 2**
>
> Thanks  for your kind suggestion.
> Sparse regularization (SR) is a regularization method for probabilities, independent of labels and distinct from a loss function. We tend to compare with robust loss functions in this paper.
> Following the suggestion, we compared our method with SR. The results for 0.8 symmetric and real-world noise are provided.
>
> |Method|CIFAR-10|CIFAR-100|WebVision|
> |---|---|---|---|
> |CE+SR|51.13±0.51|17.35±0.13|69.12|
> |CE$_\epsilon$+MAE|**58.96±0.70**|**26.30±0.46**|**71.32**|
>
> As can be seen, our approach significantly outperforms SR in difficult and real-world situations. In addition, SR requires dynamic adjustment of hyperparameters at each epoch, making it difficult to train.
>
> **3. Response to Weakness 3**
>
> Thanks for your kind suggestion.
>
> --- About ablation study on $\alpha$ and $\beta$.
>
> We offer the ablation experiments. For simplicity, we fix $\alpha$ and then adjust $\beta$,  $\beta=5$ for CIFAR-10 and 1 for CIFAR-100 is used in the paper. "S" is symmetric noise, and "AS" is asymmetric noise.
>
> |CIFAR-10|Clean|S (0.2)|S (0.4)|S (0.6)|S (0.8)|AS (0.1)|AS (0.2)|AS (0.3)|AS (0.4)|
> |---|:---:|:---:|:---:|:---:|:---:|:---:|:---:|:---:|:---:|
> |$\beta=1$|90.19±0.13|87.89±0.25|84.71±0.09|76.90±0.30|44.26±0.78|89.20±0.17|86.85±0.17|82.74±0.32|75.75±0.42|
> |$\beta=5$|**91.40±0.12**|**89.29±0.10**|85.93±0.19|**79.52±0.14**|**58.96±0.70**|**90.30±0.11**|**88.62±0.18**|**85.56±0.12**|**78.91±0.25**|
> |$\beta=10$|91.31±0.21|89.08±0.19|**86.06±0.08**|77.78±3.32|43.00±3.86|90.14±0.11|88.58±0.39|83.42±4.61|72.87±0.55|
> |**CIFAR-100**|**Clean**|**S (0.2)**|**S (0.4)**|**S (0.6)**|**S (0.8)**|**AS (0.1)**|**AS (0.2)**|**AS (0.3)**|**AS (0.4)**|
> |$\beta=0.5$|**71.01±0.05**|64.31±0.32|52.07±0.76|43.32±0.76|15.42±0.88|65.50±0.84|62.22±0.24|53.27±0.31|41.48±0.32|
> |$\beta=1$|70.83±0.18 |**65.45±0.31**|**59.20±0.42**|**48.15±0.79**|**26.30±0.46**|**67.58±0.04**|**64.52±0.18**|**58.47±0.12**|**48.51±0.36**|
> |$\beta=5$|67.87±0.88|60.05±0.84|51.19±1.62|26.93±1.80|10.30±2.83|63.22±0.46|51.67±0.66|40.34±5.45|25.69±0.77|
>
> --- About the better trade-off
>
> The better trade-off means that we achieve better performance on both fitting ability and robustness compared to previous works.
> Previous works like GCE and SCE increase fitting ability but reduce robustness due to the CE term. In contrast, our combination retains robustness, as demonstrated by Lemma 2, and also improves fitting ability. To verify this, we present comparison results on CIFAR-10 symmetric noise:
>
> |CIFAR-10|Clean|0.2|0.4|0.6|0.8|
> |:---:|:---:|:---:|:---:|:---:|:---:|
> |GCE|89.42±0.21|86.87±0.06|82.24±0.25|68.43±0.26|25.82±1.03|
> |SCE|91.30±0.08|87.58±0.05|79.47±0.48|59.14±0.07|25.88±0.49|
> |CE$_\epsilon$+MAE|**91.40±0.12**|**89.29±0.10**|**85.93±0.19**|**79.52±0.14**|**58.96±0.70**|
>
> As can be seen, CE$_\epsilon$+MAE achieves better results on both the clean and noisy cases.
>
> **4. Response to Weakness 4**
>
> Thanks for your comment. The results for instance-dependent noise are available in the global rebuttal (please see response to Q1).
> Furthermore, we provide the excess risk bound for instance-dependent noise:
> $$\mathcal{R} _L(f  _\eta^*)\le 2\delta + \frac{2c\delta}{a},$$
>
> where $c = \mathbb{E} _\mathcal D\left(1-\eta _{\mathbf{x}}\right)$, $a=\min _{\mathbf x,k}(1-\eta _y-\eta _{\mathbf x,k})$,  $f^* _\eta$ and $f^*$ denote the global minimum of $\mathcal R_L^\eta(f)$ and $\mathcal R_L(f)$, respectively.
>
> The proof is similar to that for asymmetric noise. We will add the proof in the revised version, as there is not enough space.
>
> **5. Response to Question 1**
>
> Thanks for your kind comment. The answer is yes.
> We rename the $\alpha, \beta$ for CE$ _\epsilon$+MAE loss as  $1-\lambda$ and $\lambda$ for a better presentation, where $0 < 1 - \lambda \le 1$.
>
> For $L=$ CE$_ \epsilon$+MAE, label is $\mathbf q$, and $f(\mathbf x)=\epsilon$-softmax$ \circ h(\mathbf x)$, we have
> $L = \sum _{k=1}^K q _k(-(1-\lambda))\log f(\mathbf x) _k + \lambda(1-f(\mathbf x) _k)).$ By Lagrangian multiplier, we have
>
> $$\min _{f(\mathbf x)} \max _{\alpha, \beta _k \mathcal \geq 0} \sum _{k=1}^K q _k \left( -(1-\lambda) \log f(\mathbf x) _k + \lambda (1-f(\mathbf x) _k) \right) + \alpha \left( \sum _{k=1}^K f(\mathbf x) _k - 1 \right) - \sum _{k=1}^K \beta_k f(\mathbf x)_k$$
>
> Consider the stationary condition for $f(\mathbf x)$, we have $\frac{1}{f(\mathbf x)_k^*} + \frac{\lambda}{1-\lambda} = \frac{\alpha - \beta_k}{q_k(1-\lambda)}, 0<f(\mathbf x)_k <1$. By complementary slackness, we can get $\beta_k^* =0$ and $\alpha >0$. Hence, we have $\left( \frac{1}{f(\mathbf x)_k^*} + \frac{\lambda}{1-\lambda} \right) \propto \frac{1}{q_k}$ and consequently $f(\mathbf x)^*$ is rank consistent with $\mathbf q$, i.e., $L$ is All-$k$ calibrated. Since $L$ is non-negative, so $L$ is All-$k$ consistency.

---

> > ### Comment · Reviewer_NkMU · 2024-08-10
> >
> > Thank you for the detailed response. The authors have thoroughly addressed my questions and concerns. I will maintain my original rating.

---

> > > ### Author Response · Authors · 2024-08-11
> > > **Thanks for your feedback**
> > >
> > > Dear Reviewer NkMU
> > >
> > > Thanks for your feedback! We are pleased to address your concerns and greatly appreciate your reviews, which play a crucial role in improving our work.
> > >
> > > Best regards,
> > >
> > > The authors

---

### Official Review · Reviewer_d71F · 2024-07-10

**Soundness:** 4
**Presentation:** 4
**Contribution:** 4
**Rating:** 7
**Confidence:** 3

**Summary:**

The paper introduces “ϵ-softmax,” a method to adjust softmax outputs for better approximation to one-hot vectors, thereby mitigating the impact of label noise in classification tasks. The approach modifies the softmax layer outputs to include a controllable error term ϵ, aiming to improve noise robustness without extensive alteration to the network architecture. The authors provide theoretical backing for the effectiveness of ϵ-softmax in achieving noise-tolerant learning across various loss functions. Extensive experiments with both synthetic and real-world noisy datasets are conducted to validate the claims.

**Strengths:**

1.ϵ-softmax is presented as a plug-and-play module compatible with any existing classifier that uses a softmax layer, enhancing its practical utility.

2.The paper proves that ϵ-softmax can achieve a controlled approximation to one-hot vectors, which is significant for learning with noisy labels.

3.The methodology is backed by extensive experimental results showing its superiority over existing methods in handling label noise, with detailed results across multiple datasets and noise configurations.

**Weaknesses:**

This paper should pay attention to the axis labels of its figures. In Figure 1, the x-label is Epoch and y-label is Test Accuracy. In Figures 2 and 3, the axis labels are missing.

**Questions:**

1.This paper seems to focus on classification tasks. Does ε-Softmax also work well for regression tasks?

2.Is ε-Softmax computationally efficient in terms of training time compared with baseline methods?

3.This paper shows the excellent performance of ε-Softmax for label noise. Does ε-Softmax work for input (features) noise as well?

4.I assume that in this paper, for the label noise models, both training and testing labels are noisy. I am curious about the performance when the training labels are clean and the testing labels are noisy.

**Limitations:**

The authors have adequately addressed the limitations.

---

> ### Author Rebuttal · Authors · 2024-08-06
>
> Thanks very much for your positive comments. We would like to offer the following responses to your concerns.
>
> **1. Response to Weakness**
>
> Thanks for your kind comment. For Figures 2 and 3, we extract the high-dimensional features of the test set at the second-to-last fully connected layer, then project them into 2D embeddings using t-SNE. The x-label and y-label represent the first and second dimensions of the 2D embeddings, respectively. We will add the x-label and y-label in Figures 2 and 3 in the revised version.
>
>
> **2. Response to Question 1**
>
> Thanks for your kind comment. For regression tasks, the goal is to predict a continuous numerical output rather than a categorical probability. Therefore, the softmax function and our $\epsilon$-softmax are not suitable for regression tasks.
>
> **3. Response to Question 2**
>
> Thanks for your insightful comment. We evaluate the training time of different loss functions on CIFAR100 using ResNet34, and record the average training time of an epoch on NVIDIA RTX 4090. The results are summarized in the table below.
>
> |   Loss   | Train time (s) |
> |:--------:|:--------------:|
> | CE  |      10.17|
> | GCE |      10.24|
> | SCE |      10.29|
> | NCE+MAE |10.39  |
> | NCE+AGCE|10.46  |
> | LDR-KL |10.38   |
> |CE$_\epsilon$|10.26|
> |CE$_\epsilon$+MAE|10.38|
>
> As can be seen, the training time required for all methods is almost the same.
>
> **4. Response to Question 3**
>
> Thanks for your insightful comment.  We use random masking, Gaussian blur and solarisation to  synthesize symmetric feature noise. The experimental results with 3 random trials are as follows:
>
> | CIFAR-10 | 0.2 | 0.4 | 0.6 | 0.8 |
> |---|---|---|---|---|
> | CE |       90.76±0.21 | 89.42±0.30 | 85.86±0.29 | 77.80±0.49 |
> | GCE |      88.78±0.25 | 86.52±0.05 | 82.12±0.36 | 71.37±0.29 |
> | NCE+RCE |  89.45±0.30 | 85.93±0.08 | 79.11±0.39 | 61.06±1.09 |
> | NCE+AGCE | 89.26±0.03 | 85.22±0.10 | 77.53±0.51 | 57.76±1.61 |
> | LDR-KL |   90.17±0.10 | 87.50±0.20 | 82.62±0.19 | 71.22±0.30 |
> |CE$_\epsilon$+MAE|**91.41±0.29**|**90.27±0.15**|**87.30±0.04**|**80.03±0.23**|
> | **CIFAR-100**|**0.2** | **0.4** | **0.6** | **0.8** |
> | CE |       69.82±0.36 | 66.29±0.17 | 63.47±0.39 | 57.69±1.11 |
> | GCE |      67.49±0.79 | 65.82±0.19 | 59.70±0.54 | 48.35±0.86 |
> | NCE+RCE |  62.63±0.54 | 52.96±0.40 | 38.42±0.21 | 23.73±1.01 |
> | NCE+AGCE | 63.94±0.40 | 55.66±0.50 | 43.13±0.41 | 26.68±0.36 |
> | LDR-KL |**70.47±0.32**|**67.83±0.67**| 63.61±0.34 | 57.93±0.27 |
> |CE$_\epsilon$+MAE|70.12±0.27 | 67.11±0.20 |**64.06±0.56**|**58.34±0.56**|
>
> We believe that the feature noise primarily tests the fitting ability of the loss function, and we can see that our CE$_\epsilon$+MAE also performs very well for  feature noise.
>
>
> **5. Response to Question 4**
>
> Thanks for your kind comment. In the area of learning with noisy labels, it is commonly assumed that the training labels are noisy and the test labels are clean, which aligns with real-world applications of machine learning models. Having clean training labels and noisy test labels doesn't make sense in practical scenarios, and we don't recommend considering this scenario.

---

> > ### Comment · Reviewer_d71F · 2024-08-10
> > **Response to Rebuttal**
> >
> > Thank you for the response. The authors have addressed my concerns and I do not have other questions. I will keep my rating at 7.

---

> > > ### Author Response · Authors · 2024-08-10
> > > **Thanks for your feedback**
> > >
> > > Dear Reviewer d71F,
> > >
> > > Thanks for your feedback! We are pleased to address your concerns and greatly appreciate your reviews, which play a crucial role in improving our work.
> > >
> > > Best regards,
> > >
> > > The authors

---

### Official Review · Reviewer_5cpw · 2024-07-12

**Soundness:** 3
**Presentation:** 3
**Contribution:** 2
**Rating:** 5
**Confidence:** 4

**Summary:**

The author proposes the epsilon-softmax technique as a method to address label noise. Epsilon-softmax facilitates peaky predictions by increasing the value of the highest prediction, and it also functions to reduce the magnitude of the gradient when the prediction aligns with the given label. The author introduces the concept of All-k consistency to interpret this paradigm and presents experiments on prominent real-world benchmark datasets in the field of label noise learning, specifically WebVision and Clothing1M.

**Strengths:**

The proposed epsilon softmax by the author takes a different approach compared to existing symmetric-like functions, which aim to reduce the gradient value for entirely incorrect predictions. From my understanding, the author’s approach reduces the gradient value for predictions that match the given label. By providing the value and interpretation of this novel approach, the author has significantly broadened the scope of the label noise learning field with their straightforward yet impactful idea. The proposed method has the advantage of simple gradient computation without requiring additional high-cost operations, making it applicapable to other Label Noise Learning (LNL) methods. The author demonstrates the experimental value of this approach by applying it to both the cross-entropy loss function and the focal loss function.

**Weaknesses:**

This section addresses two major concerns. For minor concerns, please refer to the "Questions" part.
1. The author mentions in line 40 the necessity for a method that can achieve both effective learning and robustness. While the proposed method offers a different perspective compared to symmetric-like loss methods, it is challenging to assert that it fully meets this necessity. Ironically, to balance the trade-off between effective learning and robustness, the author combines CE_(epsilon) loss and MAE loss. This ability to manage trade-offs is also found in other symmetric-like loss-based methods. In this context, I am interested in understanding why the proposed method might offer a better trade-off and whether it truly provides a better trade-off. I attempted to verify this through experimental comparison, but several issues arose: (1) There are no experiments that allow for a comparison of trade-offs. Experiments demonstrating the trade-off by varying alpha and beta are necessary. (2) The performance of existing methods is reported to be lower. For example, refer to the SCE paper.
2. The proposed method introduces two additional hyperparameters: m and alpha / beta. Unfortunately, based on the recorded experimental results, the proposed method appears to be sensitive to these hyperparameters. If this is not true, providing comprehensive experimental results that show the effects of varying these hyperparameters would enhance the perceived value of the proposed method.
And if the proposed method is indeed sensitive to changes in hyperparameters, I would like to see evidence that it is not sensitive to hyperparameter variations within the in-distribution domain. I recommend performing validation and test processes to identify optimal hyperparameters (refer to the processes outlined in the GJS and JS papers).

**Questions:**

1. Based on the gradient analysis, it appears that when m is sufficiently large, the sensitivity of performance to m would not be significant. However, as shown in Table 3, the difference in performance is quite notable. Does the author have an explanation for this discrepancy? Additionally, has the author investigated the results when m is infinite, meaning the CE loss is not used when the prediction matches the label?
2. Has the author ever checked the results of using only the CE_(epslion) loss function? Sections 3.1 to 3.3 and Lemma 2 suggest the importance of the single CE_(epslion) loss function.

**Limitations:**

-

---

> ### Author Rebuttal · Authors · 2024-08-06
>
> Thanks very much for your positive comments. We would like to offer the following responses to your concerns.
>
> **1. Response to Weakness 1**
>
> Thanks for your insightful comments.
>
> --- About the better trade-off
>
> The better trade-off means that we achieve better performance on both fitting ability and robustness compared to previous works.
> Previous works like GCE and SCE [1] increase fitting ability but reduce robustness due to the CE term. In contrast, our combination retains robustness, as demonstrated by Lemma 2, and also improves fitting ability. To verify this, we present comparison results on CIFAR-10 symmetric noise:
>
> |CIFAR-10|Clean|0.2|0.4|0.6|0.8|
> |:---:|:---:|:---:|:---:|:---:|:---:|
> |GCE|89.42±0.21|86.87±0.06|82.24±0.25|68.43±0.26|25.82±1.03|
> |SCE|91.30±0.08|87.58±0.05|79.47±0.48|59.14±0.07|25.88±0.49|
> |CE$_\epsilon$+MAE|**91.40±0.12**|**89.29±0.10**|**85.93±0.19**|**79.52±0.14**|**58.96±0.70**|
>
> As can be seen, CE$_\epsilon$+MAE achieves better results on both the clean and noisy cases.
>
> --- About issue (1)
>
> We offer the ablation experiments for $\alpha$ and $\beta$. For simplicity, we fix $\alpha$ and then adjust $\beta$. $\beta=5$ for CIFAR-10 and 1 for CIFAR-100 is used in the paper. "S" is symmetric noise, and "AS" is asymmetric noise.
>
> |CIFAR-10|Clean|S (0.2)|S (0.4)|S (0.6)|S (0.8)|AS (0.1)|AS (0.2)|AS (0.3)|AS (0.4)|
> |---|:---:|:---:|:---:|:---:|:---:|:---:|:---:|:---:|:---:|
> |$\beta=1$|90.19±0.13|87.89±0.25|84.71±0.09|76.90±0.30|44.26±0.78|89.20±0.17|86.85±0.17|82.74±0.32|75.75±0.42|
> |$\beta=5$|**91.40±0.12**|**89.29±0.10**|85.93±0.19|**79.52±0.14**|**58.96±0.70**|**90.30±0.11**|**88.62±0.18**|**85.56±0.12**|**78.91±0.25**|
> |$\beta=10$|91.31±0.21|89.08±0.19|**86.06±0.08**|77.78±3.32|43.00±3.86|90.14±0.11|88.58±0.39|83.42±4.61|72.87±0.55|
> |**CIFAR-100**|**Clean**|**S (0.2)**|**S (0.4)**|**S (0.6)**|**S (0.8)**|**AS (0.1)**|**AS (0.2)**|**AS (0.3)**|**AS (0.4)**|
> |$\beta=0.5$|**71.01±0.05**|64.31±0.32|52.07±0.76|43.32±0.76|15.42±0.88|65.50±0.84|62.22±0.24|53.27±0.31|41.48±0.32|
> |$\beta=1$|70.83±0.18 |**65.45±0.31**|**59.20±0.42**|**48.15±0.79**|**26.30±0.46**|**67.58±0.04**|**64.52±0.18**|**58.47±0.12**|**48.51±0.36**|
> |$\beta=5$|67.87±0.88|60.05±0.84|51.19±1.62|26.93±1.80|10.30±2.83|63.22±0.46|51.67±0.66|40.34±5.45|25.69±0.77|
>
> --- About issue (2)
>
> Our experimental setup precisely follows that of previous work [2], ensuring a completely fair comparison. The results we reproduced are consistent with [2]. We note that the key difference between our results and those in SCE paper [1] lies in the recording of epoch accuracy. [1] recorded the best epoch accuracy, while we recorded the last epoch accuracy. Consequently, [1] avoided overfitting noisy labels in the later stage of training and reported better results. For example, for CIFAR-10 with 0.8 symmetric noise, CE achieved about 39\% accuracy at the best epoch and about 18\% at the last epoch.
>
> **2. Response to Weakness 2**
>
> Thanks for your kind comment. Ours, and most robust loss functions, are indeed somewhat sensitive to hyperparameters. For instance, NCE+AGCE [2] involves four hyperparameters.
>
> As suggested and in line with [3], we provide a more extensive hyperparameter search for learning rate and weight decay. The results are available in the global rebuttal (please see response to Q2). As can be seen, our method achieves better results in many cases.
>
> **3. Response to Question 1**
>
> Thanks for your insightful comment. Due to the excellent fitting ability of the neural network, even if $m$ is about $1e4$, the gradient will still play a guiding role in optimization.
> According to your suggestion, we conducted experiments on sufficiently large and infinity $m$. The results on CIFAR-100 symmetric noise are as follows:
>
> | CIFAR-100 | 0.2 | 0.4 | 0.6 | 0.8 |
> |---|:---:|:---:|:---:|:---:|
> | m=1e10 | 53.63±0.41 |36.94±0.16|24.56±1.31|11.24±0.46|
> | m=1e15 | 53.45±0.53 |36.54±0.47|24.51±0.71|11.42±0.53|
> | m=1e20 | 53.64±0.94 |36.75±0.01|25.47±0.60|10.95±0.17|
> |m=$\infty$|54.07±0.59|36.54±1.44|24.94±0.98|11.12±0.76|
>
> As can be seen, when $m$ is sufficiently large, there is no obvious
>  difference between the results.
>
> **4. Response to Question 2**
>
> Thanks for your kind comment. As suggested, the results of  CE$_\epsilon$ are offered as follows, where "S" is symmetric noise and "AS" is asymmetric noise:
>
> | CIFAR-10 | Clean | S (0.2) | S (0.4) | S (0.6) | S (0.8) | AS (0.1) | AS (0.2) | AS (0.3) | AS (0.4) |
> |:---:|:---:|:---:|:---:|:---:|:---:|:---:|:---:|:---:|:---:|
> | CE | 90.50±0.35  | 75.47±0.27  | 58.46±0.21  | 39.16±0.50 | 18.95±0.38| 86.98±0.31|83.82±0.04 | 79.35±0.66 | 75.28±0.58  |
> |CE$_\epsilon$|88.10±0.06 |**86.88±0.18**|**83.51±0.39**|**74.68±0.40**|**29.54±0.89**|**88.06±0.30**|**85.57±0.15**|**80.82±0.41**|**76.74±0.22**|
> |**CIFAR-100**|**Clean**|**S (0.2)**|**S (0.4)**|**S (0.6)**|**S (0.8)**|**AS (0.1)**| **AS (0.2)**|**AS (0.3)**|**AS (0.4)**|
> | CE | 70.79±0.58 | 56.21±2.04 | 39.31±0.74 | 22.38±0.74 | 7.33±0.10 | 65.10±0.74 | 58.26±0.31 |49.99±0.54|41.15±1.04 |
> |CE$_\epsilon$|69.28±0.09 |**64.35±0.20**|**58.21±0.24**|**40.27±1.62**|**19.32±1.09**|**65.89±0.12**|**60.11±0.13**|**52.74±0.11**|**42.05±1.72**|
>
> [1] Symmetric cross entropy for robust learning with noisy labels, CVPR, 2019.
>
> [2] Asymmetric loss functions for noise-tolerant learning: Theory and applications, TPAMI, 2023.
>
> [3] Generalized jensen-shannon divergence loss for learning with noisy labels, NeurIPS, 2021.

---

> > ### Comment · Reviewer_5cpw · 2024-08-12
> >
> > Thank you for your additional interpretation and experimentation. My main concerns have been largely addressed. The answer to Q1 provides a good guideline regarding the size of m, and the answer to Q2 presents results that reflect a fundamental extension of the epsilon-softmax method. Integrating these points into the main text will likely enrich the overall content. I will maintain my approval decision.

---

> > > ### Author Response · Authors · 2024-08-13
> > > **Thanks for your feedback**
> > >
> > > Dear Reviewer  5cpw,
> > >
> > > Thanks for approving our responses! We will  incorporate the additional experimental results in the final version. Your guidance has been instrumental in enhancing the quality of our work.
> > >
> > > Best regards,
> > >
> > > The authors

---

### Author Rebuttal · Authors · 2024-08-07

We appreciate all reviewers for their valuable time and insightful comments. We have carefully considered the suggestions and will revise our manuscript accordingly.  We conducted some additional experiments in this global rebuttal space. Responses to other specific comments can be found in the individual rebuttal sections for each reviewer.

**Q1: Evaluation on Instance-Dependent Noise**

Following the previous work [1], we generate noise and set noise rates for instance-dependent noise. We use the same experimental settings and parameters as those described in our paper. The best results are highlighted in bold.

| CIFAR-10 IDN | 0.2 | 0.4 | 0.6 |
|---|:---:|:---:|:---:|
| CE | 75.05±0.31 | 57.27±0.96 | 37.62±0.02 |
| GCE| 86.95±0.38 | 79.35±0.30 | 52.30±0.12 |
| SCE| 86.79±0.17 | 74.56±0.49 | 49.63±0.14 |
| NCE+RCE | 89.06±0.31 | 85.07±0.17 | 70.45±0.26 |
| NCE+AGCE | 88.90±0.22 |85.16±0.26| 72.68±0.21 |
| LDR-KL | 88.99±0.15 | 84.10±0.24 | 63.11±0.23 |
|CE$_\epsilon$+MAE|**89.27±0.42**|**85.26±0.29**|**74.32±0.89**|
| **CIFAR-100 IDN**|**0.2**|**0.4**|**0.6**|
| CE | 54.46±1.73 | 40.81±0.25 | 25.57±0.03 |
| GCE | 61.95±1.37 | 56.99±0.42 | 44.19±0.36 |
| SCE | 55.58±0.74 | 39.71±0.39 | 25.63±0.76 |
| NCE+RCE | 64.13±0.49 | 57.15±0.24 | 43.22±2.31 |
| NCE+AGCE | 65.33±0.18 | 58.59±0.68 | 43.42±0.24 |
| LDR-KL | 59.19±0.34 | 43.74±0.12 | 26.10±0.16 |
|CE$_\epsilon$+MAE|**67.44±0.19**|**60.80±0.20**|**46.53±0.54**|

As can be seen, our method achieves the best performance among compared methods for instance-dependent noise.


**Q2: More Hyperparameter Search**

We conduct more hyperparameter search about learning rate and weight decay for our method. We search for learning rates (lr) in [0.01, 0.1] and weight decays (wd) in [1e-5, 1e-4, 5e-4]. lr=0.01, wd=1e-4 for CIFAR-10 and lr=0.1, wd=1e-5 for CIFAR-100 are used in the paper. The best results are highlighted in bold.


|CIFAR-10|Clean|S(0.2)|S (0.4)|S (0.6)|S (0.8)|AS (0.1)|AS (0.2)|AS (0.3)|AS (0.4)|
|:---:|:---:|:---:|:---:|:---:|:---:|:---:|:---:|:---:|:---:|
|lr=0.01, wd=1e-5|90.81±0.29|89.05±0.32|85.59±0.54|80.24±0.43|57.20±0.23|90.03±0.08|88.24±0.13|85.38±0.07|74.82±3.79|
|**lr=0.01, wd=1e-4**|91.40±0.12|89.29±0.10|85.93±0.19|79.52±0.14|**58.96±0.70**|90.30±0.11|88.62±0.18|85.56±0.12|78.91±0.25|
|lr=0.01, wd=5e-4|**91.94±0.18**|**89.76±0.08**|86.01±0.30|78.87±0.18|56.54±0.71|90.68±0.21|88.76±0.36|85.02±0.37|77.78±0.29|
|lr=0.1, wd=1e-5|61.47±1.08|57.09±5.70|50.60±10.72|35.75±1.47|27.88±10.13|80.38±3.68|74.63±7.77|76.36±3.62|59.72±10.09|
|lr=0.1, wd=1e-4|89.00±0.49|87.96±0.31|80.03±4.28|58.34±1.87|42.41±2.65|89.28±0.18|88.27±0.23|86.61±0.30|74.49±0.30|
|lr=0.1, wd=5e-4|91.11±0.36|89.44±0.29|**86.77±0.16**|**80.47±0.87**|35.56±5.59|**91.06±0.13**|**89.58±0.29**|**87.78±0.23**|**82.47±0.56**|
|**CIFAR-100**|**Clean**|**S (0.2)**|**S (0.4)**|**S (0.6)**|**S (0.8)**|**AS (0.1)**|**AS (0.2)**|**AS (0.3)**|**AS (0.4)**|
|lr=0.01, wd=1e-5|67.62±0.50|61.21±0.23|53.00±0.01|42.29±1.25|20.03±0.93|64.62±0.71|61.69±0.13|55.10±0.52|41.98±1.12|
|lr=0.01, wd=1e-4|69.31±0.49|62.66±0.19|54.69±0.35|43.67±0.22|18.49±0.64|65.13±0.28|63.05±0.30|55.45±0.33|41.52±1.27|
|lr=0.01, wd=5e-4|72.89±0.11|64.62±0.24|55.81±0.11|37.43±1.01|10.57±0.57|70.08±0.13|64.85±0.18|50.25±0.71|31.05±0.42|
|**lr=0.1, wd=1e-5**|70.83±0.18|65.45±0.31|59.20±0.42|48.15±0.79|**26.30±0.46**|67.58±0.04|64.52±0.18|**58.47±0.12**|**48.51±0.36**|
|lr=0.1, wd=1e-4|73.63±0.12|66.89±0.22|59.69±0.42|**50.69±0.25**|13.53±1.28|70.62±0.26|**66.70±0.20**|57.97±0.60|45.78±0.61|
|lr=0.1, wd=5e-4|**75.62±0.23**|**70.96±0.20**|**64.22±0.50**|14.07±0.51|1.10±0.00|**72.86±0.06**| 52.45±0.26|23.77±1.95|12.93±1.41|

As can be seen, we achieve better results in many cases.




[1] Part-dependent Label Noise: Towards Instance-dependent Label Noise, NeurIPS, 2020.

---

### Comment · Area_Chair_C6zW · 2024-08-11

Dear Reviewers,

The deadline of reviewer-authors discussion is approaching. If you have not done so already, please check the rebuttal and provide your response at your earliest convenience.

Best wishes,

AC

---

### Decision · Program_Chairs · 2024-09-25

**Decision:**

Accept (poster)

**Comment:**

All reviewers provided positive recommendations (including 3 borderline accepts) for this paper. All reviewer recognized that the proposed method is simple and effective. They also suggested several aspects where the paper needs improvement, including hyperparameter sensitivity analysis, comparison with related works, and some ablation studies. In particular, Reviewer 4jLX pointed out the math should be rigorously defined and explained, which should be improved in the final version. Additionally, it might be better for the authors to add comparison or discussion with those most-related methods that transform the labels, logits or softmax outputs, such as label smoothing [1] and logit clipping [2].

[1] Lukasik, Michal, et al. "Does label smoothing mitigate label noise?." International Conference on Machine Learning. PMLR, 2020.

[2] Wei, Hongxin, et al. "Mitigating memorization of noisy labels by clipping the model prediction." International Conference on Machine Learning. PMLR, 2023.